# Thermodynamic Feasibility of the Black Sea CH$_4$ Hydrate Replacement by CO$_2$ Hydrate

**Bjørn Kvamme** [1,2,3,4,*] and **Atanas Vasilev** [5]

1 State Key Laboratory of Oil and Gas Reservoir Geology and Exploitation, Southwest Petroleum University, Xindu Road No. 8, Chengdu 610500, China
2 State Key Laboratory of Natural Gas Hydrate, Sun Palace South Street No. 6, Beijing 100027, China
3 Hyzenenergy, 26701 Quail Creek, Laguna Hills, CA 92656, USA
4 Strategic Carbon LLC, 7625 Rancho Vista BLVD W, Corpus Christi, TX 78414, USA
5 Institute of Oceanology—Bulgarian Academy of Sciences, First May 40, PO Box 152, 9000 Varna, Bulgaria
* Correspondence: bkvamme@strategic-carbonllc.com; Tel.: +47-934-51-95-6

**Abstract:** There is an international consensus that reductions of CO$_2$ emissions are needed in order to reduce global warming. So far, underground aquifer storage of CO$_2$ is the only commercially active option, and it has been so since 1996, when STAOIL started injecting a million tons of CO$_2$ per year into the Utsira formation. Storage of CO$_2$ in the form of solid hydrate is another option that is safer. Injection of CO$_2$ into CH$_4$ hydrate-filled sediments can lead to an exchange in which the in situ CH$_4$ hydrate dissociates and releases CH$_4$. Two types of additives are needed, however, to make this exchange feasible. The primary objective of the first additive is related to hydrodynamics and the need to increase injection gas permeability relative to injection of pure CO$_2$. This type of additive is typically added in amounts resulting in concentration ranges of additive in the order of tens of percentages of CO$_2$/additive mixture. These additives will, therefore, have impact on the thermodynamic properties of the CO$_2$ in the mixture. A second additive is added in order to reduce the blocking of pores by new hydrates created from the injection gas and free pore water. The second additive is a surfactant and is normally added in ppm amounts to the gas mixture. A typical choice for the first additive has been N$_2$. The simple reasons for that are the substantial change in rheological properties for the injection gas mixture and a limited, but still significant, stabilization of the small cavities of structure I. There are, however, thermodynamic limitations related to adding N$_2$ to the CO$_2$. In this work, we discuss a systematic and consistent method for the evaluation of the feasibility of CO$_2$ injection into CH$_4$ hydrate-filled reservoirs. The method consists of four thermodynamic criterions derived from the first and second laws of thermodynamics. An important goal is that utilization of this method can save money in experimental planning by avoiding the design of CO$_2$ injection mixtures that are not expected to work based on fundamental thermodynamic principles. The scheme is applied to hydrates in the Black Sea. Without compositional information and the knowledge that there is some verified H$_2$S in some sites, we illustrate that the observed bottom hydrate stability limits are all with hydrate stability limits of hydrates containing from 0 to 3 mole% H$_2$S. A limited number of different injection gas mixtures has been examined, and the optimum injection gas composition of 70 mole% CO$_2$, 20 mole% N$_2$, 5 mole% CH$_4$, and 5 mole% C$_2$H$_6$ is feasible. In addition, a surfactant mixture is needed to reduce blocking hydrate films from injection gas hydrate.

**Keywords:** hydrate; thermodynamics; CH$_4$/CO$_2$ swapping; phase transitions; thermodynamic properties

## 1. Introduction

The latest report [1] of the Intergovernmental Panel on Climate Change discusses five possible future scenarios. Different timelines of human climate action determine the future,

from better to apocalyptic [2]. In the most optimistic scenario, the world cuts emissions immediately and avoids the worst catastrophe, but the climate continues to warm for at least 30 years due to the existence of greenhouse gases in the atmosphere. Solomon et al. estimate the long-term climate effects of emitted carbon dioxide ($CO_2$) remaining for several hundreds to thousands of years [3]. A key task to avoid the increasing impacts of global warming is $CO_2$ storage. The possibility to capture $CO_2$ as a solid $CO_2$ hydrate in the robust marine gas hydrate reservoirs, with simultaneous production of methane, is a win-win situation. Coupled with steam, cracking of the produced methane opens up a possible cycle with only hydrogen ($H_2$) as the externally delivered product, a very environmentally friendly and future-oriented concept. For marine gas hydrate (GH) deposits (GHDs), the volume of methane hydrate is the first percentage of the volume of sediments between the seafloor and the bottom simulating reflector (BSR) (for the Black Sea GHDs in average ~2%; see Section 2 for more details). Therefore, the bigger part of hydrate reservoirs is empty or with methane hydrate saturations without commercial interest and ready for $CO_2$ storage as hydrate.

Injection of $CO_2$ into methane ($CH_4$) hydrates is a safe option for long-term, offshore, underground storage. There are several reasons for this statement. Sealing (shale, clay) integrity has been verified through millions of years of trapping the in situ natural gas hydrate. The thermodynamic reason is that hydrate stability depends on the temperatures, pressures, and concentrations (associated chemical potentials) of all components in all phases. Despite this trivial thermodynamic fact, it is quite common to refer to the pressure and temperature projection of this multi-dimensional dependency as the hydrate equilibrium curve and the only hydrate stability limit to be considered. Infinite dilution chemical potentials for guest molecules in water (and seawater) are very low and lower than what is possible for a guest molecule trapped in a hydrate cavity [4]. Exposing hydrate to pure water or seawater containing almost zero guest molecules will, therefore, lead to hydrate dissociation [4].

Fractures that lead to the transport of seawater into the hydrate will, therefore, result in the dissociation of in situ $CH_4$ hydrate due to the lower chemical potential of $CH_4$ in the seawater (infinite dilution of $CH_4$). There are numerous scientific papers documenting natural gas seeps to the oceans from hydrate-filled sediments. Depending on the conditions of temperature and pressure on the seafloor, this may occur as release of free gas into the water column above or though formation of hydrates on the seafloor. Examples of the latter can be found in, for instance [5,6]. The hydrate mounds formed on the seafloor are not thermodynamically stable because the hydrocarbon content in the surrounding seawater is very low. The corresponding hydrocarbon chemical potentials in the seawater are lower than the chemical potentials of the same hydrocarbons in the hydrate, and the hydrate will dissociate. The constant release of hydrocarbons from this dynamic hydrate formation and dissociation leads to biological ecosystems, ranging from microorganisms to biological species that can be visualized [7,8]. If the temperature on the seafloor is too high and/or the pressure too low for hydrates from the released gas to exist on the seafloor, then the gas distributes into the ocean. Depending on the leakage gas flux, this can form nano bubbles to macro-scale bubbles [7,8], and even sometimes large gas plumes.

Trapping the $CO_2$ into a solid structure is a second benefit. A third benefit is that the potential dissociation of $CO_2$ hydrate due to surrounding pore water will lead to higher-density water that will sink.

$CO_2/CH_4$ swapping in natural gas hydrate-filled sediments is a concept that will have to compete with aquifer storage of $CO_2$. The Norwegian company EQUINOR has aquifer storage of $CO_2$ as a business area. $CO_2$ injection into aquifers is a standard procedure that has been proven by several years of daily practice. Since 1996, a million tons of $CO_2$ per year, separated from the Sleipner hydrocarbon system in the North Sea, has been injected into the aquifers in the Utsira formation [9–12]. It remains unverified whether there are $CO_2$ leakages in the ocean or not. It is also unknown how far the $CO_2$ plume has been moving and whether it has crossed the border of the Norwegian Exclusive Economic Zone (EEZ) or not. To increase the injection permeability, some $CH_4$ is added to the $CO_2$ to get the system closer to the critical point at injection conditions. Injecting $CO_2$ into $CH_4$ hydrate-filled sediments involves a lower liquid water fraction of the pores as compared to a corresponding aquifer reservoir with the same pore characteristics. The addition of nitrogen or air to $CO_2$ is one way to increase injection gas permeability. One challenge is that the mole-fraction $N_2$ in the $CO_2/N_2$ mixture has historically been based on trial and error rather than thermodynamic evaluation procedures.

If the primary goal is to benefit from a win-win situation of $CH_4$ release from gas hydrate, there is a need for a kinetically efficient mechanism. One such mechanism is to ensure that the injection gas can create a new gas hydrate that releases enough heat to dissociate in situ $CH_4$ hydrate. This was one of the challenges related to the Ignik Sikumi gas hydrate production test on the North Slope of Alaska in 2012 [13,14]. The injection gas was simply too dilute in $N_2$ to be able to create a new hydrate with free water in the pores [14]. Given this, we proposed a new systematic scheme for the evaluation of injection gas feasibility for combined, safe, long-term storage and energy production in the form of $CH_4$ release [15]. $CO_2$ undergoes a phase transition to a higher density at a temperature slightly above 284.14 K. This leads to a jump in hydrate stability pressures for $CO_2$ hydrate and hydrates created from $CO_2/N_2$ mixtures. There are many misunderstandings related to these shifts in the limits of hydrate existence regions versus similar limits for $CH_4$ hydrate. The problem is that temperature and pressure are independent thermodynamic variables in the first law and the combined first and second laws of thermodynamics. A projection of hydrate stability limits in temperature and pressure is not a measure of hydrate stability. Hydrate stability is a function of the combined first and second laws of thermodynamics.

The primary purpose of this paper is to provide a deeper thermodynamic explanation of the four criterions proposed by Kvamme and Vasilev [15], and as such, hopefully make it easier for other researchers in this topic to apply the same scheme in their studies. There are two important differences between this paper and the one submitted by Kvamme and Vasilev [15]. This paper goes a bit deeper into the thermodynamic laws and explains how they impact reasonable choices for mixtures of injection gas intended for the combined production of hydrocarbons and safe long-term storage of $CO_2$ as a hydrate. More specifically, this paper also goes into more detail on a criterion that was used by Kvamme [14] in the evaluation of the Ignik Sikumi pilot. The unconditional stability of hydrates also requires that all gradients of Gibbs free energy also would lead to hydrate stability (negative free energy change). One of these is the chemical potential of water, which for conditions at Ignik Sikumi, would not be fulfilled for higher mole% $N_2$ in $CO_2/N_2$ than roughly 25 mole%. Yet another difference from the paper submitted by Kvamme and Vasilev [15] is that we examine a substantially broader range of Black Sea hydrates. These are regions within the EEZs of Bulgaria, Romania, Russia, and Turkey. It is of special interest to check the thermodynamic approach in this work (and [15]) against the conclusion of Schicks et al. [16], namely, "the injection of $CO_2$ or a $CO_2$–$N_2$ gas mixture is not applicable for the Danube Paleodelta in the Black Sea, because the local pressure and temperature conditions are too close to the equilibrium conditions of both, the $CO_2$ hydrate and a $CO_2$–$N_2$ mixed hydrate stability fields".

A second goal is to illustrate how different injection mixtures are expected to perform in terms of $CH_4/CO_2$ swapping over extended regions of general temperature and pressure regions.

The third goal of this paper is to illustrate the procedure for some specific conditions of natural gas hydrates in the Black Sea.

To our knowledge, we are the only one that utilizes residual thermodynamics for all phases, including hydrate. In a thermodynamic paper like this, we have not found any relevant papers to refer to beyond our papers on residual thermodynamics. This does not mean that there are not many theoretical papers in the open literature. Quite the opposite—there are very many good papers, but not within the scope of this paper. References to open literature are, therefore, limited to experimental papers for model verifications and to papers with comparable reference states (residual thermodynamics) also for hydrate. The lack of references to experimental papers also includes our experimental papers during the latest 3 decades. This is a thermodynamic analysis paper, and experimental data are merely utilized (and referenced appropriately) for model verifications.

The innovation in these two papers, this one and the one submitted by Kvamme and Vasilev [15], is that they, for the first time, utilize a systematic method for analysing the feasibility of $CO_2/CH_4$ swapping based on the fundamental thermodynamic laws. This includes two criteria based on the combined first and second laws. Criterion 1 is related to hydrate stability limits, criterion 2 (a) is related to hydrate Gibbs free energy, and criterion 2 (b) is related to gradients of hydrate Gibbs free energy. Criterion 3 is directly related to a kinetically efficient mechanism of heat released from the formation of new hydrate from injection gas as a heat source for the dissociation of in situ hydrates. This criterion is related to the first law of thermodynamics. Criterion 4 is related to the second law and directly addresses the level of temperature needed for breaking hydrogen bonds efficiently.

This work is not a review paper, since this paper and the submission by Kvamme and Vasilev [15] are fundamentally new and the first work to go back to thermodynamic laws. Analysis along the lines presented in these papers can save money in avoiding experiments, which would otherwise be wasted. Unfortunately, the spending related to the Ignik Sikumi [13,14] also could have been spent much more efficiently if a thermodynamic analysis along the lines of that here and in Kvamme and Vasilev [15] had been applied.

Another very important aspect of these papers, this one and Kvamme and Vasilev [15], is the fact that there are no actual technology challenges related to the concept of aquifer storage of $CO_2$, which has been operative since 1996 [9–12]. As will be discussed also in more detail, the kinetics of $CO_2/CH_4$ swapping is not slow at all. The formation of a new hydrate from injection gas is instant on the macro level of time. Blocking hydrate films will, however, delay massive $CO_2/CH_4$ swapping. This is also discussed in even more detail by Kvamme and Vasilev [15]. Some illustrations from experiments on this are also included in that paper [15].

The novelty of this method is that it is unique, general, and based on fundamental thermodynamics, and it uses a completely uniform reference system for the analysis. It is the only method for screening $CO_2$ injection gas mixtures based on a thermodynamic model, which can be used to compare different hydrates in terms of the first law, the second law, and the combined first and second law. While the title contains a very specific region, the method is totally general for any hydrate resources worldwide. The method is a breakthrough that can save money in the planning of $CO_2/CH_4$ swapping projects. A number of reported experimental studies have failed—for obvious reasons, according to the method here—and could have been avoided. In terms of costs, the most important failure was the Ignik Sikumi pilot test in Alaska [13,14]. Injection gas mixture will always form structure I hydrate. The in situ hydrate may be purely structure I hydrate, which is quite frequent, or a combination of structure I and II hydrates. For the layer case, the injection gas mixture has to be designed based on the most stable in situ hydrate.

The paper is organized as follows. Section 2, "Materials and Methods", outlines the data used as an example in the feasibility analysis. The feasibility analysis method and thermodynamic criterions are also derived in this section. Examples from various regions of the Black Sea are discussed in Section 3, "Results". The paper is closed with discussion and conclusions in Sections 4 and 5, respectively.

## 2. Materials and Methods

The primary method in this work is classical thermodynamics utilizing the combined first and second laws of thermodynamics, the first law of thermodynamics, and the second laws of thermodynamics. Details of the method are briefly outlined in Section 2.2 to Section 2.4. A summary of the feasibility analysis method is provided in Section 2.4.3. Since the purpose of the method is a thermodynamics analysis aiming towards feasible solutions for safe storage of $CO_2$, then the materials in this work are relevant sites for $CO_2$ storage as hydrate and their characteristics. We limit ourselves to the Black Sea, and materials are discussed in Section 2.1

### 2.1. Black Sea Gas Hydrates

The aim of this section is:

- to present the available data about the Black Sea GHDs. Today, they are determined only with geophysical methods, mainly from BSRs on seismic records and positive resistivity anomalies registered with Controlled Source Electromagnetics (CSEM);
- to estimate the parameters needed for the first estimation of the Black Sea potential for $CO_2$ storage in GHDs:
  1. the areas of the GHDs and the sediment volumes under these areas between the seabed and the BSR surface (the base of the GHSZ);
  2. the average temperature and pressure in sediments at the depth of the BSR as extreme values of the main independent thermodynamic parameters;
  3. the volumes of GHs and methane.

The sources of the metadata and interpretation results are the 15 publications in the *Marine and Petroleum Geology* special issue "Black Sea Gas Hydrates", 2020–2021 (Editors: M. Haeckel, G. Bohrmann, K. Schwalenberg, W. Kuhs, and K. Wallmann) [17] and cited in the publications. Geophysical, geochemical, drilling, etc. data are the result of the cruises of about 2 dozen international projects acknowledged in the above publications. The last detailed review of available evidence about hydrate occurrences in Europe was published by the participants in the EC COST Action MIGRATE in 2020 [18]. "World Atlas of Submarine Gas Hydrates in Continental Margins", 2022 (Editors: J. Mienert, C. Berndt, A. M. Tréhu, A. Camerlenghi, and C. Liu) [19] presents recent Black Sea information, too.

Figure 1 shows recently updated data about the Black Sea gas hydrates and related objects (see Figure 1 caption for details) as gas seepages; mud volcanos, Cretaceous volcanos, earthquakes; and heat flow stations. Main basin ridges, highs, sub-basins, fans, and troughs are shown, too.

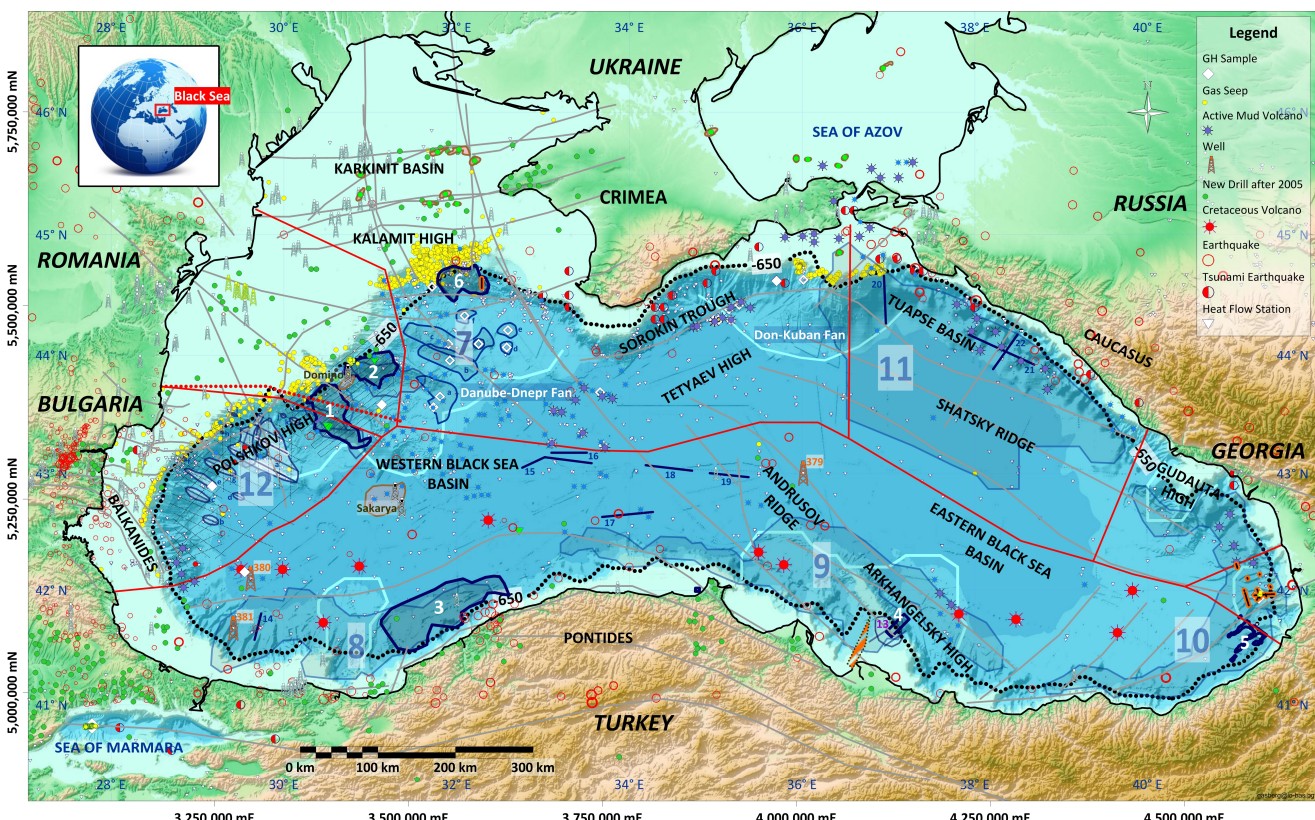

**Figure 1.** Black Sea gas hydrates and related objects (References in Table 1). Black Sea gas hydrate stability zone (GHSZ) lateral boundary for methane hydrates (approximated with the isobath 650 m); 6 GHDs (numbered BSR areas; the description in the text); 6 areas and 10 2D seismic lines with BSRs; 34 stations with GHs in the sediment samples; 54 faults; 2091 gas seepages; 52 active mud volcanos; 121 stations with mud breeches in the sediment samples (potential mud volcanos not proved with geophysical methods); 11 Cretaceous volcanos; 746 earthquakes (M 2-6); 36 historical earthquakes with tsunami; heat flow stations: 3 DSDP (Deep Sea Drilling Project), 4 Ifremer with 70 m long gravity corer, 2378 with heat flow probes; 7 EEZs; 15 BG drills and production platform Galata; 7 RO platforms; 216 drills after 2005; 2 largest gas fields: Sakarya, TR, 405 bcm and Domino, RO, 80 bcm; seismic lines from 2 cruises: MSM34 in the BG and RO EEZs, SUGAR project; Halliburton GS 1992 in the BG EEZ. According to Tari and Simmons [20]; 2 basin ridges (Andrusov and Shatsky), 5 highs (Polshkov, Kalamit, Tetyaev, Arkhangelsky, and Gudauta), 2 sub-basins (Karkinit and Tuapse), 4 fans (Danube Dnepr and Don-Kuban), and Sorokin Trough are shown (All data available upon request).

Estimated BSR areas or GHDs are divided into 3 groups. The most studied is the first group (COST Action MIGRATE; Figure 1: numbers 1–6). The second group is from national publications and the public EMODnet database (Figure 1: 7–12, 7a–f, 12a–k). The level of study in groups 1 and 2 decreases with the increase of their numbers. The third group consists of the parts of seismic records with registered BSRs (Figure 1: 14–22).

BSR area 13 is out of the methane GHSZ and could be explained with 10% hydrogen sulfide ($H_2S$) in methane hydrates [21]. To estimate lines as areas, we assume that the lines with BSR are axes of areas with BSR with a width of 2 km (approximate lines width in Figure 1).

**Table 1.** Black Sea GHDs (BSR areas) ordered by EEZs. $H_W$—average water depth; $H_{BSR}$—average thickness of the GHSZ (sediment volume between the seafloor and the BSR) determined from Figures 2 and 3; $V_{GHSZ}$—sediment volume between the seafloor and the BSR; $V_{GH}$—volume of GHs; $V_{CH4}$—the volume of methane (1 bcm = 1 km$^3$); $T_{BSR}$—average temperature on the BSR boundary (average GHD temperature $T_{GHD} = (T_{BSR} + 9.1)/2$); $P_{BSR}$—average pressure on the BSR boundary. No 13 is assumed with a 10% content of $H_2S$ [21] and is not included in TR Total. GHDs are at 2 research levels—Initial and Less and the Less level GHDs values are shown in *Italic* font.

| State Country Code | No | Area km² | $H_W$ m | $H_{BSR}$ M | $V_{GHSZ}$ km³ | $V_{GH}$ km³ | $V_{CH4}$ bcm | G mK/m | $T_{BSR}$ °C | $P_{BSR}$ bar | References ** |
|---|---|---|---|---|---|---|---|---|---|---|---|
| **BG** | **1** | 3006 | 1200 | 330 | 992 | 18 | 2768 | 26 | 18 | 153 | [17] |
|  | *12 \** | *518* | *1600* | *250* | *129* | *2* | *361* | *44* | *20* | *185* | [18] |
| **Total** |  | **3524** |  | **318** | **1121** | **20** | **3129** |  |  |  |  |
| **GE** | *10(1/2) \** | *1287* | *1300* | *130* | *167* | *3* | *467* | *42* | *15* | *143* | [19,21] |
| **RO** | **2** | 1849 | 900 | 220 | 407 | 7 | 1135 | 24 | 14 | 112 | [13] |
| **RU** | *11 \** | *6290* | *1800* | *340* | *2138* | *38* | *5966* | *40* | *23* | *214* | [19,21] |
|  | *20–22 \** | *42* | *1600* | *230* | *10* | *0* | *27* | *48* | *20* | *183* | [20] |
| **Total** |  | **6332** |  | **339** | **2148** | **39** | **5993** |  |  |  |  |
| **TR** | **3** | 7080 | 1700 | 260 | 1841 | 33 | 5136 | 43 | 20 | 196 | [13] |
|  | **4** | 324 | 740 | 100 | 32 | 1 | 90 | 46 | 14 | 84 | [13] |
|  | **5** | 482 | 1300 | 120 | 58 | 1 | 161 | 53 | 15 | 142 | [13] |
|  | *8 \** | *2349* | *1300* | *230* | *540* | *10* | *1507* | *52* | *21* | *153* | [21] |
|  | *9 \** | *3534* | *1400* | *260* | *919* | *17* | *2563* | *40* | *20* | *166* | [21] |
|  | *10(1/2) \** | *1287* | *1400* | *190* | *244* | *4* | *682* | *42* | *17* | *159* | [19,21] |
|  | *14–19 \** | *70* | *2000* | *300* | *21* | *0* | *59* | *35* | *20* | *230* | [20] |
| **Total** |  | **15,125** |  | **242** | **3656** | **66** | **10,199** |  |  |  |  |
| **UA** | **6** | 1950 | 900 | 140 | 273 | 5 | 762 | 40 | 15 | 104 | [13] |
|  | *7 \** | *687* | *1600* | *340* | *234* | *4* | *652* | *28* | *19* | *194* | [22] |
| **Total** |  | **2637** |  | **192** | **507** | **9** | **1413** |  |  |  |  |

\* Assumed 10% of the area in Figure 1 with BSR. ** Additional information in [23–27].

Table 2 presents the general and minimum-maximum data about the Black Sea GHDs from Table 1, grouped according to the level of their research.

**Table 2.** Main physical parameters of the Black Sea GHDs (BSR areas).

| Research Level | Area km² | $H_W$ min/max M | $H_{BSR}$ M | $V^s_{GHSZ}$ km³ | $V_{GH}$ km³ | $V_{CH4}$ Bcm | G min/max mK/m | $T_{BSR}$ min/max °C | $P_{BSR}$ min/max Bar |
|---|---|---|---|---|---|---|---|---|---|
| Initial | 14,691 | 740/1700 | 245 | 3603 | 65 | 10,052 | 24/53 | 14/20 | 84/196 |
| *Less* | *16,063* | *1300/2000* | *274* | *4403* | *79* | *12,284* | *28/52* | *15/21* | *143/230* |
| **Total** | **30,753** |  | **260** | **8006** | **144** | **22,336** |  |  |  |

In Table 1 the values of $H_W$, $H_{BSR}$, and G are determined from the model parameters of the Black Sea GHSZ [28] shown in Figures 2–4.

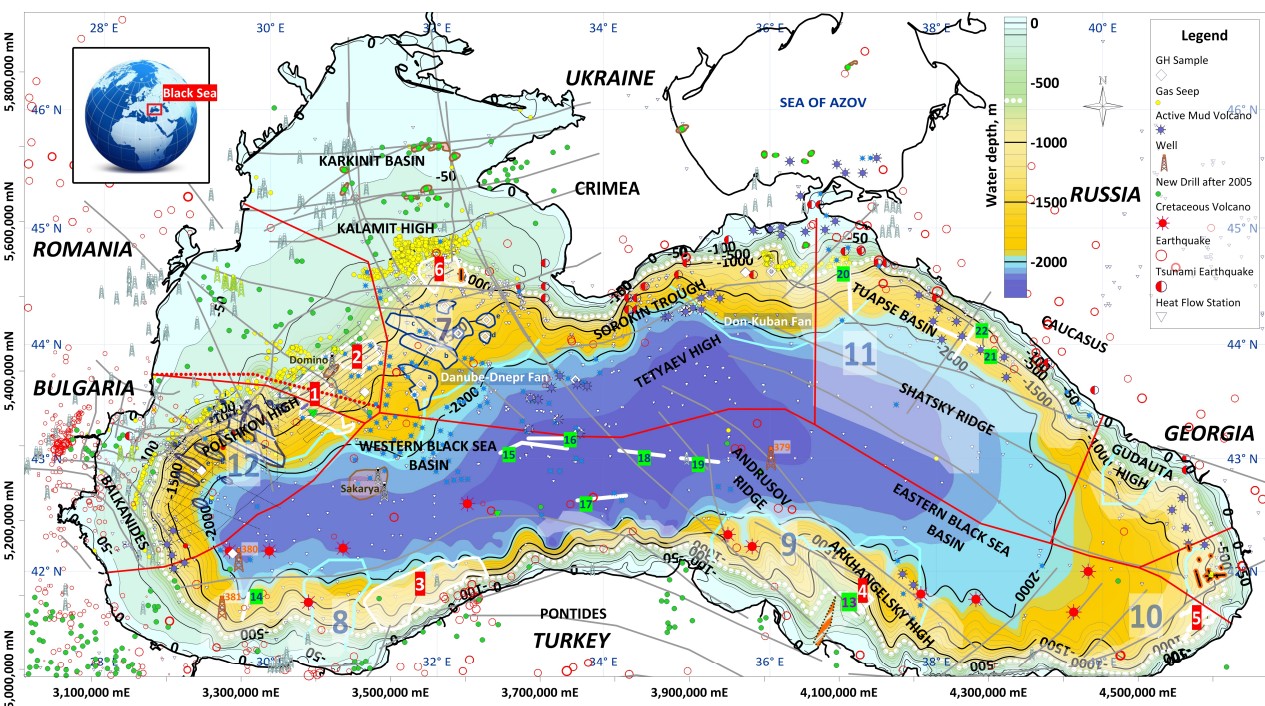

**Figure 2.** Black Sea bathymetry [23]. Data and objects from Figure 1.

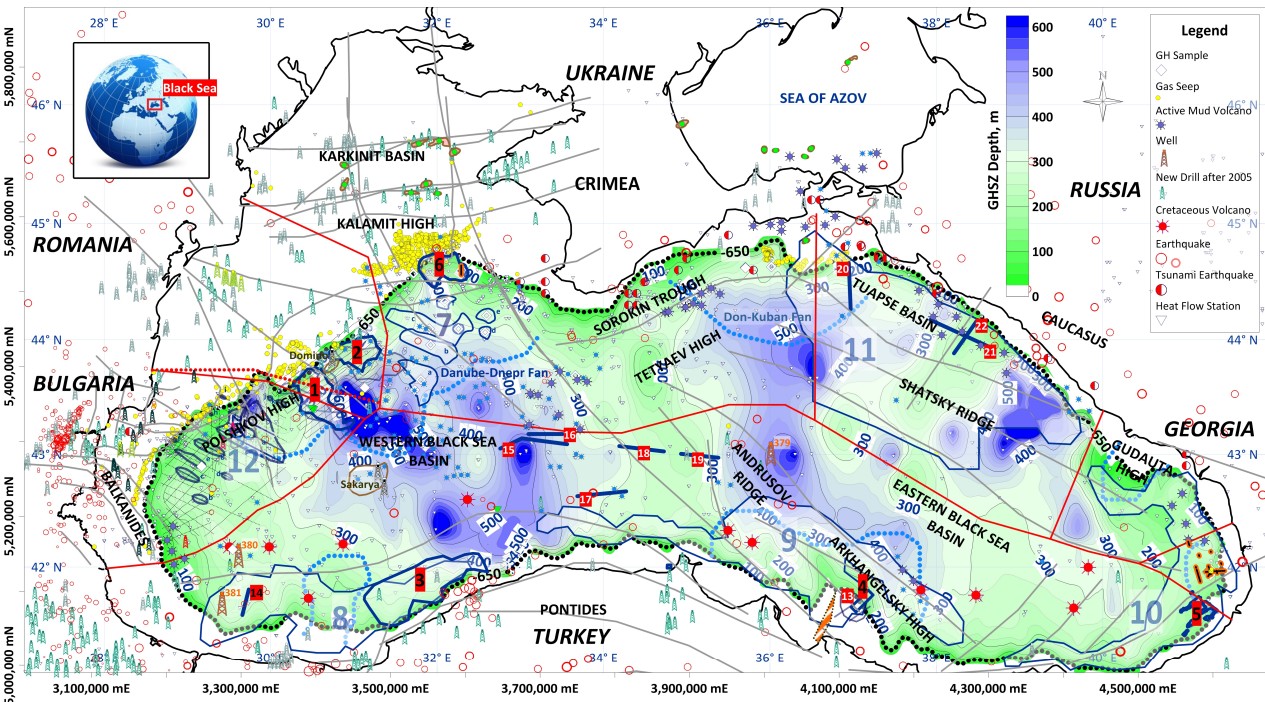

**Figure 3.** Black Sea GHSZ thickness [28]. Data and objects from Figure 1.

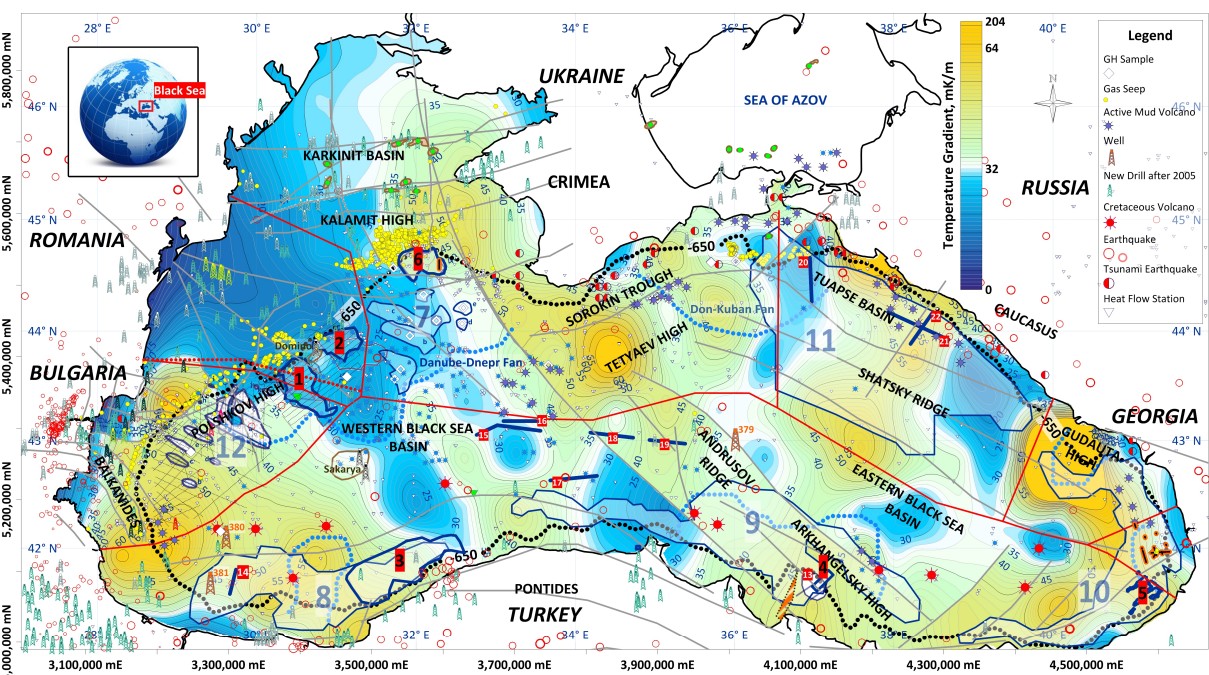

**Figure 4.** Black Sea geothermal gradient [28]. Data and objects from Figure 1.

### 2.2. Hydrate Stability Limits

As mentioned in the text above, some points contain much $H_2S$ (even up to 10 mole%) and are excluded from the tables of the data considered in this work. This is, however, an indication that there might be mixtures of thermogenic and biogenic sources of hydrocarbons for the in situ hydrates. Practically, this implies that the in situ hydrates can be mixtures of structure I and structure II hydrates. Without very definite composition data, we might check the sensitivity of $CH_4$ hydrate relative to $CH_4$ containing $H_2S$. Even small amounts of $H_2S$ have a substantial impact on hydrate temperature pressure stability limits [29,30]. Without detailed information on the composition of the hydrocarbon mixtures, we might as well use the structure I hydrate with pure $CH_4$ hydrate and the three mixtures containing $H_2S$ as a model for evaluation since all the points selected from Table 1 fall well in between the 4 solid curves in Figure 5a. Corresponding Gibbs free energies in Figure 5b illustrate the impact of small amounts of $H_2S$ on hydrate stability.

### 2.3. Scientific Methods

The scientific methods utilized in this paper are classical thermodynamics and non-equilibrium thermodynamics. A consistent residual thermodynamic scheme is applied throughout the paper. This means that ideal gas properties for all components in all phases at actual temperature and pressure is the reference state. Thermodynamic consistency is hardly a method, but it is mentioned here in order to exclude some enthalpy estimation methods based on the Clapeyron equation, and simplifications of that approach. Consistent entropy is only possible if the model for Gibbs free energy and the model for enthalpy are consistent. Since the model for enthalpy is directly derived from the Gibbs free energy model using standard thermodynamic relationships, then this is fulfilled.

Since the general model has been published in many papers referred to in this paper, some finer theoretical details are excluded in order to save space. Researchers that want to reproduce the results presented here will, therefore, benefit from also reading the original papers describing the different aspects of the model. Some of these papers also provide convenient fits of thermodynamic properties to temperature and pressure that make it even easier to adopt a residual thermodynamic method based on the model utilized in this work.

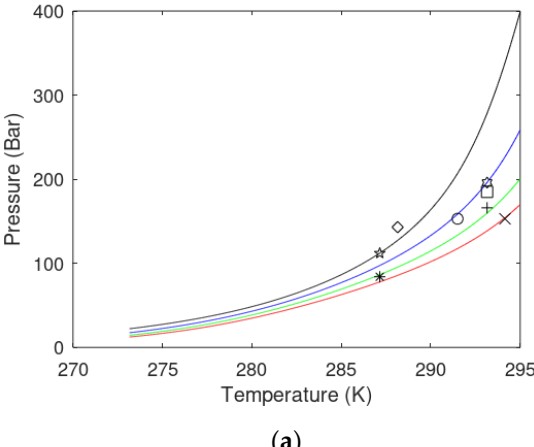
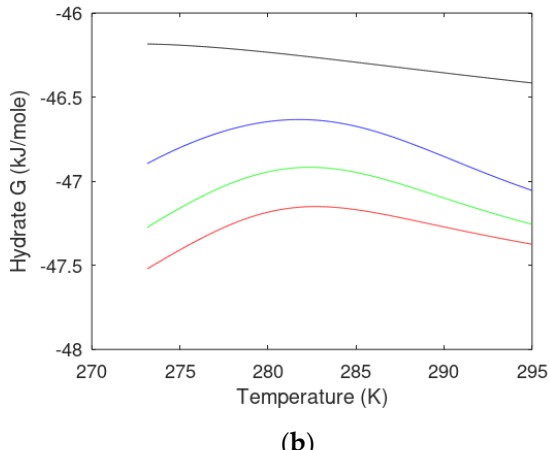

(**a**)            (**b**)

**Figure 5.** (**a**) Calculated pressure-temperature stability limits for hydrates formed from pure $CH_4$ (black), from 99 mole% $CH_4$ and 1 mole% $H_2S$ (blue), from 98 mole% $CH_4$ and 2 mole% $H_2S$ (green), and from 97 mole% $CH_4$ and 3 mole% $H_2S$ (red). Single points refer to Table 1 condition for different sites. The various symbols represent different site numbers on Table 1. The circle is 1, the square is 12, the diamond is 10, the pentagram is 2, the hexagram is 3, the star is 4, the plus is 9, and the cross is 8. Regarding temperature, 10 and 5 are almost identical in the same temperature and 1 bar higher pressure in 10 relative to 5. (**b**) Molar Gibbs free energy for hydrate from pure $CH_4$ and the 3 mixtures of $CH_4$ with $H_2S$.

### *2.4. A Classical Thermodynamic Approach for Theoretical Evaluation of $CO_2/CH_4$ Swap Feasibility*

The evaluation method proposed by Kvamme and Vasilev [15] is based on one criteria based on hydrate existence limits in temperature pressure projection. In this section, we explain what this first criteria means thermodynamically. The three other criteria are directly linked to the thermodynamic laws, specifically, the combined first and second laws, the first law, and the second law.

Section 2.4.1 outlines the thermodynamics laws, and in Section 2.4.2, we detail the four criteria [15], and thermodynamic implications are discussed in Section 2.4.3.

#### 2.4.1. Thermodynamic Laws

The first law of thermodynamics for an open flowing system is most conveniently expressed in terms of enthalpy H. The line under H denotes extensive property, and the unit is Joule. Similarly, the line under volume V is extensive volume in $m^3$. $Q^{(m)}$ is the heat supplied to phase m from external sources and other co-existing phases. m is a phase index that runs over all coexisting phases. P is pressure, $\mu$ is chemical potential in unit Joules/mole, and N is mole numbers. Component i runs over all components n in the system. The second term is denoted as shaft work. The reason is that it is the total work minus internal push work in the flowing system. This internal work cannot be exported out for use or distribution to other phases. The third term on the right-hand side is historically denoted as chemical work, although there are no chemical reactions involved in the systems discussed in this work. Chemical potential consists of two contributions. A partial molar energy accounts for the molecular interaction energies in phase m, as well as ideal gas energies. The second contribution is related to the entropy of phase m and accounts for how this is shifted with changes in pressure, temperature, and changed composition during phase transitions,

$$d\underline{H}^{total} \leq \sum_{m=1}^{phases} \left[ dQ^{(m)} + \underline{V}^{(m)} dP^{(m)} + \sum_{i=1}^{n.m} \mu_i^{(m)} dN_i^{(m)} \right] \tag{1}$$

The second law of thermodynamics for the same system can be written as:

$$dS = \sum_{m=1}^{phases} dS^{(m)} \tag{2}$$

The entropy development for every phase inside the sum of (m) is on the form of:

$$dS^{(m)} = \frac{dQ^{(m)}}{T^{m,ext}} \tag{3}$$

The superscript *m,ext* in Equation (3) involves temperatures from external sources, but also from other phases with different temperatures. The second part of the second law is that heat cannot be transported directly from a lower temperature to a higher temperature.

Entropy-related energy changes are denoted as irreversibility and frequently even denoted as "friction losses", in a wider sense of the term. Gibbs free energy, G, is the available energy when the effects of irreversibility are subtracted from the enthalpy.

$$dG^{total} \leq \sum_{m=1}^{phases} -S^{(m)}dT^{(m,surr.)} + V^{(m)}dP^{(m)} + \sum_{i=1}^{n.m} \mu_i^{(m)} dN_i^{(m)} \leq 0 \tag{4}$$

The minimum of Equation (4) does not mean that each local phase at a given time is unconditionally stable. Quite the opposite is true. For a phase to be unconditionally stable, additional constraints must be fulfilled:

$$\left[ \frac{\partial G^{(m)}}{\partial M} \right]_{K \neq M} \delta M \leq 0 \tag{5}$$

for any possible range of changes of the independent variable M.

2.4.2. Thermodynamic Criteria for $CH_4/CO_2$ Feasibility

There are two implications of Equation (4). For systems that can reach equilibrium, the inequality sign in (4) is replaced by an equal sign for reversible phase equilibrium.

Both pressure and temperature are always defined locally in hydrate-filled sediment, or during flow in a pipeline or process equipment. With 2 components (water and $CH_4$) distributed over 3 phases (water, gas, and hydrate), the Gibbs phase rule results in 1 degree of freedom. The equivalent counting behind the Gibbs phase rule is the number of independent thermodynamic variables minus constraints. Constraints are conservation laws and equilibrium equations [30–32]. Defining 2 independent variables implies that the system is over-determined. The result is, of course, the same if the compressed form of the Gibbs phase rule is utilized. A disadvantage of the Gibbs phase rule as applied to hydrate systems is that careful consideration of the number of active phases is needed. $CO_2/N_2$ mixtures will adsorb selectively on liquid water as a precursor to hydrate formation. This means that the water surface concentration of $CO_2$ will be higher than the gas concentration of $CO_2$, and the $N_2$ surface concentration is correspondingly lower. In simplified terms, selective adsorption on the liquid water surface is controlled by the different gas components' "desire" to condense out from the gas and the interactions of these molecules with liquid water molecules in the gas/water interface. Several adsorption models might be used to illustrate selective adsorption. One specific model was used by Kvamme [14]. As long as the composition of the adsorbed layer has a different composition and a different density than the other phases, it is a unique phase by thermodynamic definition. With 3 components (water, $CO_2$, and $N_2$) and 4 active phases (liquid water, adsorbed on liquid water, gas, and hydrate), the number of degrees of freedom is still 1, according to the Gibbs phase rule. The number of active phases are even higher than the phases mentioned above. Interactions between charged atoms in mineral surfaces and outside water and guest molecules results in several routes to hydrate formation. Adsorbed water

and guest molecules on the mineral surfaces can restructure into hydrate. Another example is that structured water traps guest molecules in typical distances of 3 to 4 adsorbed water layers [30]. A very important implication of a mathematically over-determined system is that the chemical potentials for the different components in different phases may not be the same [31,32]. The reason is that mass conservation laws obviously have to be fulfilled. When there is a mismatch between the number of independent thermodynamic variables and the number of constraining equations, the criteria of equal chemical potentials for the different components in different phases may not be impossible to fulfill [31,32]. It is, however, still possible to calculate hydrate stability limits in different sets of variables. The pressure temperature stability limit is calculated in the same manner as hydrate equilibrium, with the assumptions that no water enters the hydrate from the gas phase and no guest molecules enter from the dissolved state in the liquid water. Using the mass balance between the phases, then Equation (4) can be reformulated for a fixed point of temperature and pressure as:

$$
\left[\mu_{H_2O}^{(H)}(T,P,\vec{x}^H) - \mu_{H_2O}^{(aq)}(T,P,\vec{x}^{aq})\right]dN_{H_2O}^{(H)} + \left[\mu_{CO_2}^{(H)}(T,P,\vec{x}^H) - \mu_{CO_2}^{(gas)}(T,P,\vec{x}^{gas})\right]dN_{CO_2}^{(H)}
$$
$$
+ \left[\mu_{N_2}^{(H)}(T,P,\vec{x}^H) - \mu_{N_2}^{(gas)}(T,P,\vec{x}^{gas})\right]dN_{N_2}^{(H)} = 0 \tag{6}
$$

Superscript *H* denotes hydrate, *aq* is liquid water phase, and gas is the gas mixture of $CO_2$ and $N_2$. The arrow on mole-fraction x denotes the vector of mole-fractions in the specific phase denoted by the phase superscript.

There is nothing very special about Equation (6) relative to other phase transitions, but the result is presented just to illustrate that the pressure temperature hydrate stability limit does not say anything explicit about hydrate stability, but merely provides a projection of the conditions of the pressure and temperature for which a hydrate can exist. The following three equations need to be solved for the stability limits of hydrates created from $CO_2$ and $N_2$.

$$
\left[\mu_{H_2O}^{(H)}(T,P,\vec{x^H}) - \mu_{H_2O}^{(aq)}(T,P,\vec{x^{aq}})\right] = 0 \tag{7}
$$

$$
\left[\mu_{CO_2}^{(H)}(T,P,\vec{x^H}) - \mu_{CO_2}^{(gas)}(T,P,\vec{x^{gas}})\right] = 0 \tag{8}
$$

$$
\left[\mu_{N_2}^{(H)}(T,P,\vec{x^H}) - \mu_{N_2}^{(gas)}(T,P,\vec{x^{gas}})\right] = 0 \tag{9}
$$

Using pure water as an example, Equation (7) can be rewritten to:

$$
\left[\mu_{H_2O}^{O,H}(T,P) - RT(3/23)\ln\left(1 + h_{CO_2,large} + h_{N_2,large}\right) - RT(1/23)\ln\left(1 + h_{N_2,small}\right)\right]
$$
$$
-\mu_{H_2O}^{(aq)}(T,P) = 0 \tag{10}
$$

Equation (10) applies to structure I hydrate. The superscript *O,H* denotes the chemical potential in empty clathrate. Pure liquid water chemical potential, as well as empty hydrate chemical potential, are available from Kvamme and Tanaka [33]. h is the canonical partition function for the actual guest molecules in a specific cavity type, as indicated by the superscript. $CO_2$ is generally too large to enter the small cavity of structure I. There is some evidence in open literature that some $CO_2$ gets trapped in the small cavity of structure I at conditions far below the freezing point. Further, in very special, chemically stimulated systems that promote extreme solution concentrations of $CO_2$ in water, intended to promote $CO_2$ hydrate formation inside "bulk" water, $CO_2$ has been observed in the small cavity. There is no evidence that $CO_2$ has been observed in the small cavity for hydrate formed at liquid water conditions and the regular solubility of $CO_2$ in water.

$$h_{CO_2,large} = e^{\beta[\mu_{CO_2,large}^H(T,P,\vec{x}^H) - \Delta g_{CO_2,large}(T)]} \tag{11}$$

$$h_{N_2,large} = e^{\beta[\mu_{N_2,large}^H(T,P,\vec{x}^H) - \Delta g_{N_2,large}(T)]} \tag{12}$$

$$h_{N_2,small} = e^{\beta[\mu_{N_2,small}^H(T,P,\vec{x}^H) - \Delta g_{N_2,small}(T)]} \tag{13}$$

$\Delta g$ is free energy of inclusion for the specific component in the specific cavity type, as given by the subscript. From Equations (8) and (9), the chemical potential for guest molecules in gas and hydrate are the same, and the gas chemical potentials replaces the hydrate guest chemicals in Equations (11) to (13). The Soave Redlich Kwong (SRK) equation of state [34] is used for $CO_2$ and $N_2$ in Equations (8) and (9). The chemical potential for component k in the gas is trivially given by the ideal gas chemical potential, the ideal gas mixing terms from the mixing entropy, and the residual term for the fugacity coefficient $\phi_k^{(gas)}$ from SRK [34]. The $CO_2$ and $N_2$ molecules are both linear, with only one specific independent moment of inertia for each of the two molecules. Equations for the ideal gas chemical potential are available in any textbook in physical chemistry and not needed in the context of this paper. The chemical potential for a molecule k in gas is given by Equation (14) below in residual thermodynamics.

$$\mu_k^{(gas)}(T,P,\vec{x}) = \mu_k^{idealgas,pure}(T,P) + RT \ln\left[x_k^{(gas)}\phi_k^{(gas)}(T,P,\vec{x}^{(gas)})\right] \tag{14}$$

Either $T$ or $P$ must be defined, and the other independent variable is then solved iteratively from Equation (10). For $CH_4$ hydrate, Equation (15) is solved iteratively in the same manner, using the same procedure as the derivations above for the two-component gas of $CO_2$ and $N_2$.

$$\left[\mu_{H_2O}^{O,H}(T,P) - RT(3/23)\ln\left(1 + h_{CH_4,large}\right) - RT(1/23)\ln\left(1 + h_{CH_4,small}\right)\right] \\ -\mu_{H_2O}^{(aq)}(T,P) = 0 \tag{15}$$

As for the solutions of Equation (10), also the solution of Equation (15) has nothing to do with hydrate stability. The solutions simply describe in which regions of temperature and pressure the hydrates can exist. Some examples of hydrate stability limits for $CO_2/N_2$ mixtures are plotted in Figure 6a. With the same reference state for all components, the relative stability of different hydrates is given by comparisons of the Gibbs free energies for the hydrates. Specifically:

$$G_{CO_2/N_2}^H = x_{H_2O}^H \mu_{H_2O}^H + x_{CO_2}^H \mu_{CO_2}^H + x_{N_2}^H \mu_{N_2}^H \tag{16}$$

$$G_{CH_4}^H = x_{H_2O}^H \mu_{H_2O}^H + x_{CH_4}^H \mu_{CH_4}^H \tag{17}$$

The hydrate mole-fractions of guest molecules are given by the statistical mechanical model, and the calculation goes through the filling fractions of different guest molecules in different cavity types [35–38]. The Gibbs free energies for the same hydrates plotted in the temperature-pressure stability limits in 1 (a) is plotted in 1 (b).

The combined first and second laws of thermodynamics, as given by Equation (4), provide the two first criteria for $CH_4/CO_2$ swapping feasibility. Pure $CO_2$ is not feasible in terms of temperature and pressure. Further, 20 mole% $N_2$ and 30 mole% $N_2$ reduce the "impossible" range to a very limited region of temperatures and pressures. Replacing 1 mole% of $N_2$ with $CH_4$ makes both $CO_2/N_2$ mixtures feasible. This is the first criteria.

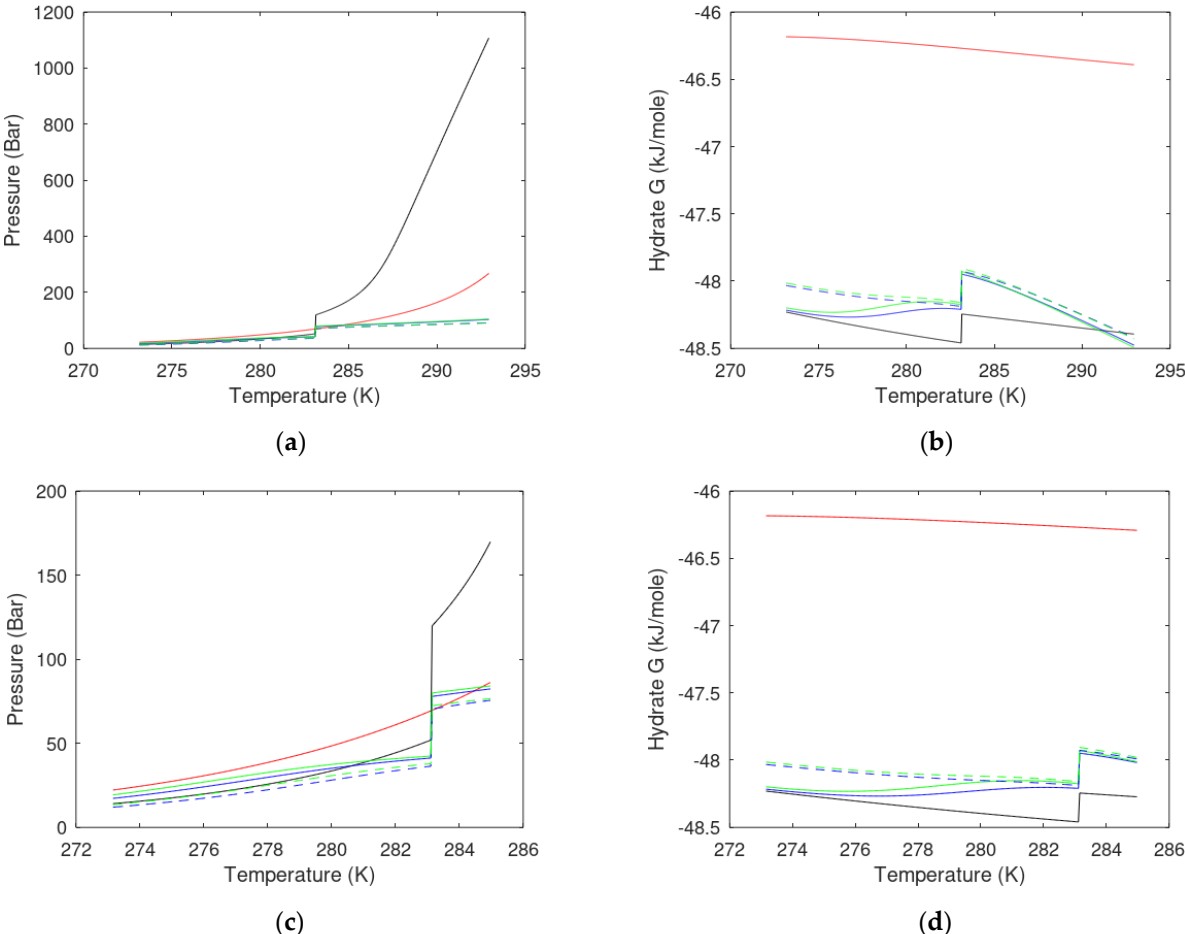

**Figure 6.** (**a**) Temperature pressure stability limits for the range of temperatures relevant for Danube region of the Black Sea. Red curve is $CH_4$ hydrate and black curve is $CO_2$ hydrate. Blue curve is for hydrate from 80 mole% $CO_2$ and 20 mole% $N_2$. Green curve is for hydrate from 70 mole% $CO_2$ and 30 mole% $N_2$. Blue dashed curve is for hydrate from 80 mole% $CO_2$, 19 mole% $N_2$, and 1 mole% $CH_4$. Green dashed curve is for hydrate from 70 mole% $CO_2$, 29 mole% $N_2$, and 1 mole% $CH_4$. (**b**) Gibbs free energy for the hydrates in Figure 6a. Figure 6c and d are more focused versions of Figure 6a and b and only contains the data for temperatures below 285 K.

The second criteria actually involves both Equations (4) and (5). All investigated $CO_2/N_2$ systems (up to 30 mole% $N_2$) have lower Gibbs free energy than the Gibbs free energy for $CH_4$ hydrate and are thermodynamically more stable.

A consequence of Equation (5), however, is that the maximum $N_2$ content is around 25 mole% $N_2$ for some conditions below the $CO_2$ phase transition point slightly above 284.14 K [14]. This does not mean that 30 mole% $N_2$ in the injection gas is not feasible since selective adsorption of $CO_2$ on the liquid water surface may lead to higher concentrations of $CO_2$ than 25 mole% on the water surface. Recent experiments [29] also confirm that injection of 30 mole% $N_2$ in the $CO_2/N_2$ mixture is feasible. Temperatures in the Black Sea are somewhat higher in relevant areas. In Figure 7, we, therefore, examine the chemical potentials for water in liquid and hydrate for different fractions of $N_2$ and 4 different pressures. Some $N_2$ is beneficial for filling small cavities, and the optimum balance of the mixtures examined, in terms of the water chemical potential, is 80 mole% $CO_2$. Water driving forces for hydrate formation is limited for 150 bars pressure, but sufficient for 200 bar and 250 bar.

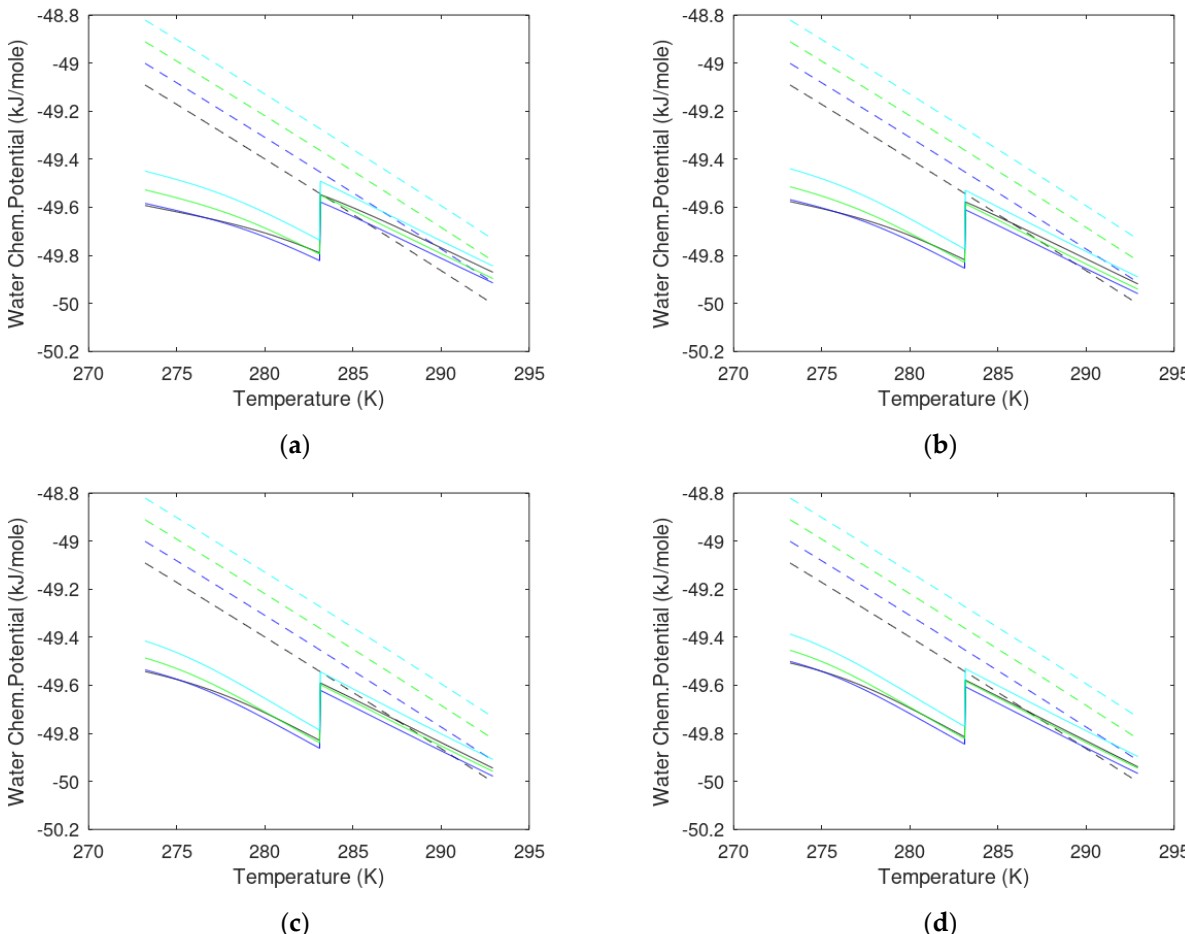

**Figure 7.** (**a**) Liquid water chemical potential (dashed) and hydrate water chemical potential (solid) for four different pressures and a $CO_2/N_2$ mixture containing 95 mole% $CO_2$. Black curve is for 100 bar pressure, blue is for 150 bar pressure, green is for 200 bar pressure, and cyan is for 250 bar pressure. (**b**) Same as (**a**) but for 90 mole% $CO_2$. (**c**) Same as (**a**) but for 80 mole% $CO_2$. (**d**) Same as (**a**) but for 70 mole% $CO_2$.

There are several ways to use the first law, but few that have sufficient information to be feasible. As such, the easiest way is to consider this as two systems connected by liquid water in between. Hydrate formation is a nano-scale process in time and space [31,32,35–39]. On a macroscopic time scale (seconds and up), formation of a new hydrate from injection gas is instant. The first law criterion is simply that the released heat from injection gas hydrate formation is sufficient to dissociate in situ $CH_4$ hydrate. Experimental data for enthalpies of hydrate formation are frequently given in units of kJ/mole guest. The enthalpy model utilized in this work was derived by Kvamme [40]. See, for instance, references [40–42] for verification of the enthalpy model through comparisons with experimental data.

Enthalpies of hydrate formation in units of kJ/mole hydrate are plotted in Figure 8a, along with mole-fractions of $CO_2$ and $CH_4$ in the hydrates in Figure 8b.

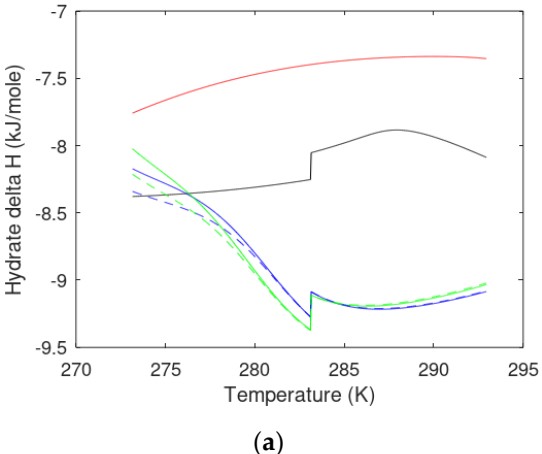
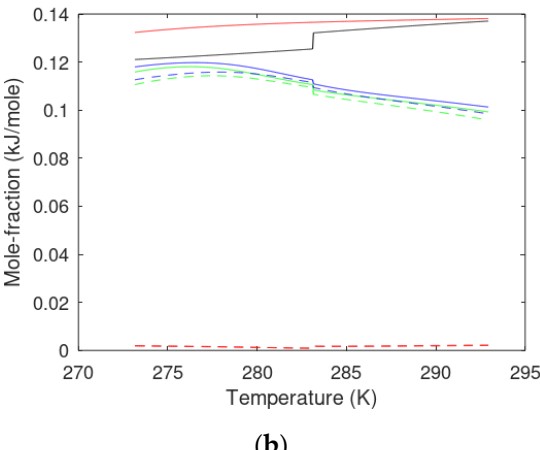

(a)　　　　　　　　　　　　　　　　　　　　　　　(b)

**Figure 8.** (**a**) Enthalpy of hydrate formation in units of kJ/mole hydrate along temperature pressure hydrate stability limits in Figure 6 (a) Red curve is $CH_4$ hydrate and black curve is $CO_2$ hydrate. Blue curve is for hydrate from 80 mole% $CO_2$ and 20 mole% $N_2$. Green curve is for hydrate from 70 mole% $CO_2$ and 30 mole% $N_2$. Blue dashed curve is for hydrate from 80 mole% $CO_2$, 19 mole% $N_2$, and 1 mole% $CH_4$. Green dashed curve is for hydrate from 70 mole% $CO_2$, 29 mole% $N_2$, and 1 mole% $CH_4$. (**b**) Mole-fractions $CO_2$ and $CH_4$ in hydrate. Red curve is $CH_4$ hydrate and black curve is $CO_2$ hydrate. Blue curve is $CO_2$ mole-fraction for hydrate from 80 mole% $CO_2$ and 20 mole% $N_2$. Green curve is for $CO_2$ mole-fraction in hydrate from 70 mole% $CO_2$ and 30 mole% $N_2$. Blue dashed curve is $CO_2$ mole-fraction for hydrate from 80 mole% $CO_2$, 19 mole% $N_2$, and 1 mole% $CH_4$. Green dashed curve is $CO_2$ mole-fraction for hydrate from 70 mole% $CO_2$, 29 mole% $N_2$, and 1 mole% $CH_4$. Upper red dashed curve is $CH_4$ mole-fraction in hydrate from 70 mole% $CO_2$, 29 mole% $N_2$, and 1 mole% $CH_4$ and lower red dashed curve is $CH_4$ mole-fraction in hydrate from 80 mole% $CO_2$, 19 mole% $N_2$, and 1 mole% $CH_4$.

We did not plot the mole-fraction $N_2$ in Figure 8 since the important issues are the amount of $CO_2$ that can be stored and the associated $CH_4$ release. Note in particular the change in $CO_2$ uptake after the increase in $CO_2$ density. The reason is the associated lower chemical potential for $CO_2$ in the gas phase, which make it more favorable for $CO_2$ to remain in gas as compared to entering hydrate. The mole-fraction $CH_4$ in hydrate for the two mixtures containing 80 mole% $CO_2$, 19 mole% $N_2$, and 1 mole% $CH_4$ and 70 mole% $CO_2$, 29 mole% $N_2$, and 1 mole% $CH_4$, respectively, are hard to distinguish in the figure. These numbers are not very important, but indicate a level of $CH_4$ mole-fraction in hydrate; for these mixtures, the mole-fractions $CH_4$ in hydrate for 273.16 K is 0.002025 and 0.001816 for the 70 mole% $CO_2$ mixture and 80 mole% $CO_2$ mixture, respectively. The corresponding numbers for 292.84 K are 0.002256 and 0.002042. The significance of the digits is not discussed, and likely only the first 4 digits are significant.

Criterion 4 is directly related to the second law, and the main implication is that the temperature has to be high enough to break hydrogen bonds in solid hydrate and the hydrate/liquid water interface, and to increase the entropy sufficiently from ordered hydrate to disordered liquid water and gas. This criterion is hard to evaluate without more details on pore geometry, hydrate saturation, and several other pore characteristics. With some higher level of information than what is available for this study, it will at least be possible to establish various levels of models. These can range from phenomenological models to different levels of theoretical models, such as, for instance, Phase Field Theory [43–48]. As mentioned above, the formation of a new $CO_2$-dominated hydrate from injection gas is "instant" on a macro level of modeling. It is a pore-scale model, and it is, therefore, expected that this criterion is fulfilled for injection of $CO_2/N_2$ gas mixtures with an additional surfactant (or surfactant mixture) to reduce mass transport-blocking hydrate films from the injection gas.

2.4.3. Summary $CH_4/CO_2$ Feasibility Evaluation Scheme

The two first criteria are related to the combined first and second laws of thermodynamics.

1.  Pressure temperature projection for hydrate stability limits of the injection gas hydrate must at least be below the hydrate stability limits for the in situ $CH_4$ hydrate for the range of temperatures and pressures relevant for the actual site and sediments section. This criterion is evaluated in a way similar to the comparison of hydrate equilibrium curves in a temperature-pressure projection.
2.  Gibbs free energy will always try to reach a minimum as a function of the temperature, pressure, and masses in the system.
3.  The Gibbs free energy of the hydrate formed from injection gas must be lower than the Gibbs free energy for the in situ $CH_4$ hydrate for the relevant range of local conditions in the real sediment.
4.  Gradients in Gibbs free energy changes must also be negative (towards lower Gibbs free energy). Practically, this implies that each component must individually benefit from entering the hydrate forming from the injection gas. In thermodynamic language, it strictly means that for each component in the new hydrate, the chemical potentials for the water and guests must be lower than the chemical potentials for the same components in the original phases. Water will dominate, and there may be cases in which fulfillment of 2 (a) and sufficient water chemical differences will dominate enough to provide efficient hydrate formation. These exceptions will leave a new hydrate under gradients of hydrate dissociation in chemical potential gradients.

The criterion from the first law is:

1.  Heat released during the formation of a new hydrate from injection gas must be sufficient to dissociate in situ $CH_4$ hydrate.
    The criterion from the second law is:
2.  The level of temperature from the formation of a new hydrate from injection gas must be sufficiently high to efficiently break the hydrogen bonds in the water/hydrate interface and in "bulk" hydrate and provide the necessary increase in entropy from a low entropy in hydrate to higher entropies in disorganized liquid water and gas phases.

## 3. Results

The stabilizing effect of $H_2S$ is very high, as illustrated by Figure 5. Designing an injection gas for this system may involve also $H_2S$ if the released gas contains $H_2S$ that has to be separated from the hydrocarbons. This $H_2S$ can then be reinjected as part of the $CO_2$ mixture injection gas. Some thermodynamic calculations on this option are conducted and discussed in Section 3.1. Another alternative is to add a hydrocarbon that will fill the large cavity of structure I. This may seem like a bad idea since the hydrate store capacity for $CO_2$ lies in the large cavities. The challenge with $CO_2$ is that the steep change in hydrate stability limits pressures for temperatures higher than the density increase phase transition at 283.14 K. The effects of adding small amounts of ethane ($C_2H_6$) are discussed in Section 3.2. These are only a few alternatives, and there might be many other injection mixtures that can do the same.

*3.1. $CO_2/CH_4$ Swap Feasibility through Adding $H_2S$*

Without detailed composition information on the hydrates in Table 1, we use the $CH_4$ hydrate and the three mixtures with $H_2S$ as a basis for swapping with injection gas. Figure 9 illustrates that just adding some $CH_4$ will not make it feasible for the $CO_2/CH_4$ swap over the entire region of temperatures and pressures.

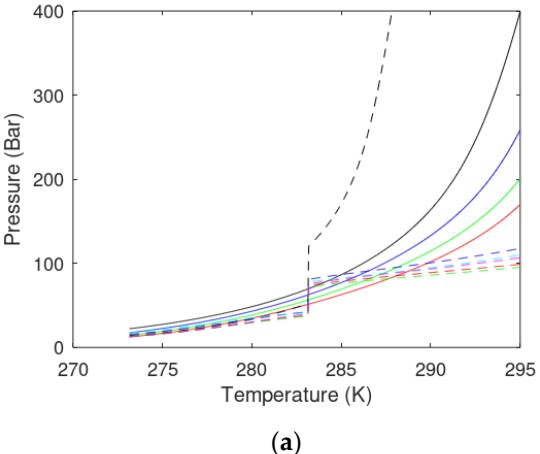
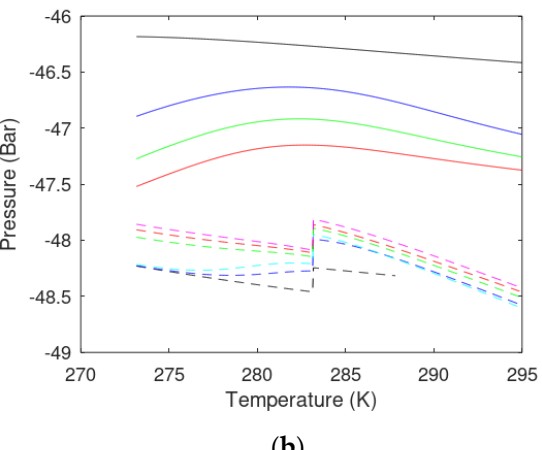

(**a**)                                                                                    (**b**)

**Figure 9.** (**a**) Calculated pressure temperature stability limits for hydrates formed from pure $CH_4$ (black), from 99 mole% $CH_4$ and 1 mole% $H_2S$ (blue), from 99 mole% $CH_4$ and 2 mole% $H_2S$ (green), and from 99 mole% $CH_4$ and 3 mole% $H_2S$ (red). Dashed curves are for different compositions of injection gas. Black is for hydrate from pure $CO_2$. Blue is for hydrate from 90% $CO_2$ and 10% $N_2$. Cyan is for hydrate from 80% $CO_2$ and 20% $N_2$. Red is for hydrate from 75% $CO_2$, 20%$N_2$, and 5% $CH_4$. Green is for hydrate from 80% $CO_2$ and 20% $N_2$. Red is for hydrate from 75% $CO_2$, 23%$N_2$, and 2% $CH_4$. Red is for hydrate from 75% $CO_2$, 15%$N_2$, and 10% $CH_4$. (**b**) Molar Gibbs free energy for hydrate from pure $CH_4$ and the 3 mixtures of $CH_4$ with $H_2S$. Same color codes as in Figure 9a.

Adding small amounts of $CH_4$ and small amounts of $H_2S$ is just an example of how the situation can be changed, as illustrated in Figure 10. Manipulating with just these components in addition to $N_2$ is of course artificial. The primary purpose of the addition gases is to increase gas permeability for injection, while at the same time not destroying $CO_2$/$CH_4$ swap possibilities. As such, $N_2$ (or air) is ideal since the effect of dilution of $CO_2$ on $CO_2$ chemical potential is partly compensated for by some $N_2$ filling in small cavities.

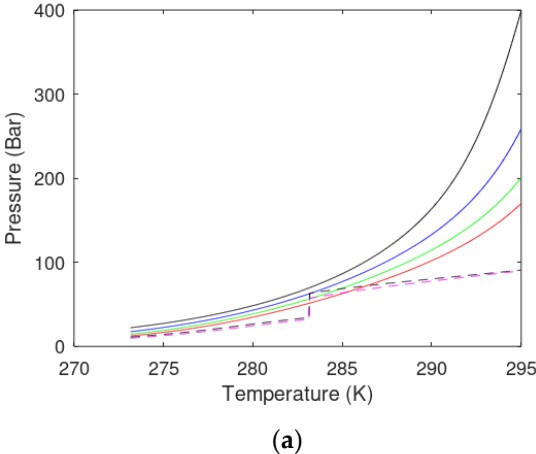
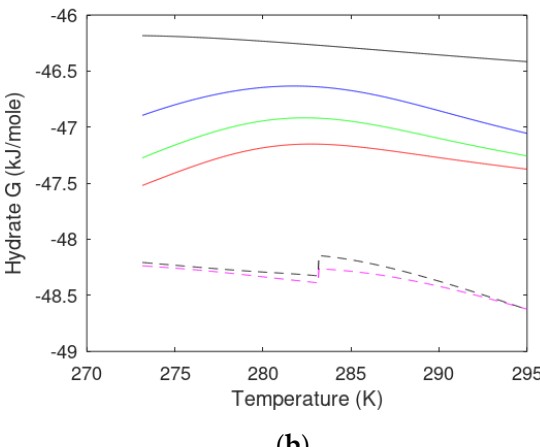

(**a**)                                                                                    (**b**)

**Figure 10.** (**a**) Calculated pressure temperature stability limits for hydrates formed from pure $CH_4$ (black), from 99 mole% $CH_4$ and 1 mole% $H_2S$ (blue), from 99 mole% $CH_4$ and 2 mole% $H_2S$ (green), and from 99 mole% $CH_4$ and 3 mole% $H_2S$ (red). Dashed curves are for different compositions of injection gas. Red is for a mixture of 75% $CO_2$, 1% $H_2S$, 1% $CH_4$, and 23% $N_2$. Magenta is for a mixture of 75% $CO_2$, 2% $H_2S$, 3% $CH_4$, and 20% $N_2$. (**b**) Molar Gibbs free energy for hydrate from pure $CH_4$ and the 3 mixtures of $CH_4$ with $H_2S$. Same color codes as in Figure 10a.

Both injection mixtures containing $H_2S$ are feasible, except for a very limited range of temperatures and pressure. In addition, keep in mind that the efficient mechanism does

not necessarily bring the in situ hydrate and hydrate from the injection gas, in contact (and competition) since released heat from formation of the new hydrate should be able to dissociate in situ hydrates. It is, therefore, expected that in situ hydrate and hydrate formed from injection gas are separated by water.

Adding $H_2S$ to injection gas is an interesting thermodynamic exercise, and the effects are visible. It is not desirable in real life. Even though criterion 1 may not be the most important, since formation of new hydrate is separated from in situ hydrate, we may add other components.

### 3.2. CO₂/CH₄ Swap Feasibility through Adding C₂H₆

The challenge with $CO_2$ and criterion 1 is the phase transition to higher density at 283.14 K. Adding small amounts of a component that can also partly fill large cavities will counteract this effect. $CH_4$ will not compete very efficiently on large cavity occupation, but $C_2H_6$ will. Adding $C_2H_6$ is, of course, slightly more expensive than adding $CH_4$, but we have to evaluate towards the sales value of $CO_2$ storage. In the context of this paper, only two compositions are illustrated in Figure 11.

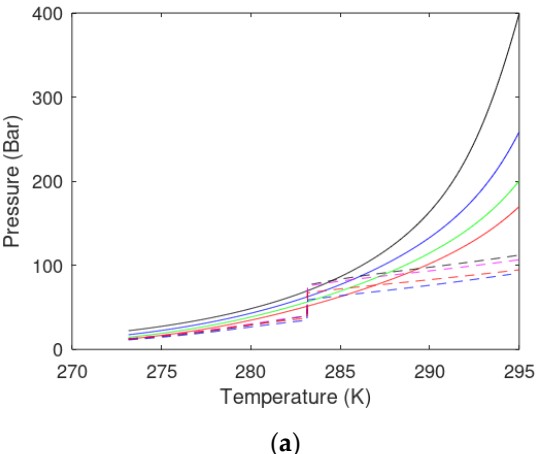 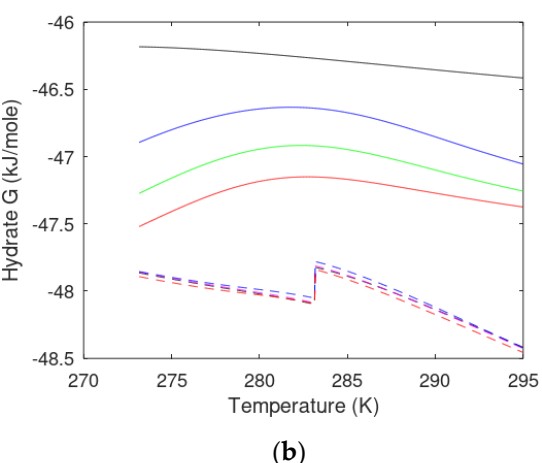

(**a**)                    (**b**)

**Figure 11.** (**a**) Calculated pressure temperature stability limits for hydrates formed from pure $CH_4$ (black), from 99 mole% $CH_4$ and 1 mole% $H_2S$ (blue), from 99 mole% $CH_4$ and 2 mole% $H_2S$ (green), and from 99 mole% $CH_4$ and 3 mole% $H_2S$ (red). Dashed curves are for different compositions of injection gas. Black is for a mixture of 80 % $CO_2$, 10% $CH_4$, and 10% $N_2$. Magenta is for a mixture of 75 % $CO_2$, 10% $CH_4$, and 15% $N_2$. Blue is for 70% $CO_2$, 5% $CH_4$, 5% $C_2H_6$, and 20% $N_2$. Red is for 70% $CO_2$, 5% $CH_4$, 2% $C_2H_6$, and 23% $N_2$. (**b**) Molar Gibbs free energy for hydrate from pure $CH_4$ and the 3 mixtures of $CH_4$ with $H_2S$. Same color codes as in Figure 11a.

Returning to criteria 3 and the energy balance related to the first law, the parameters for $H_2S$ were only derived from Molecular Dynamics simulations for temperatures up to 285 K. Extrapolation for pure $H_2S$ may be appropriate for chemical potentials and, as such, also for pressure temperature stability limits. Verification of the model calculations for these properties can be found elsewhere [49–52]. Extrapolations for pure component enthalpies of formation may also be fair, but with unknown accuracy. Yet, extrapolations of partial molar enthalpies of $H_2S$ in mixtures to the high temperatures of the Black Sea are more unclear. Rather than speculating, we can look at pure component enthalpies since, after all, $H_2S$ are in fairly small amounts in the "model" mixture illustrated in Figure 5. Enthalpies of hydrate formation are illustrated in Figure 12.

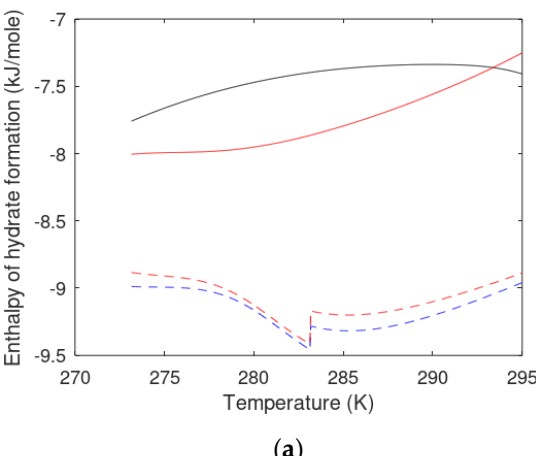
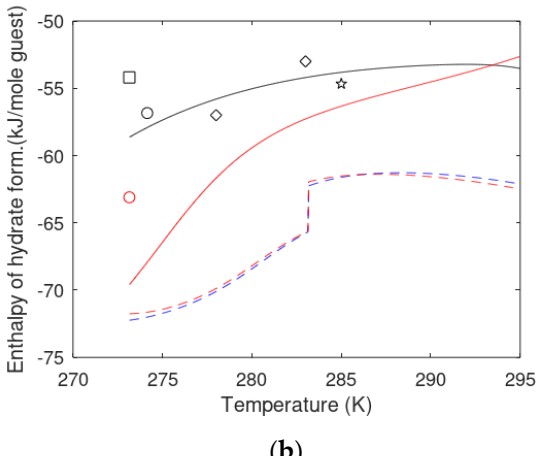

| | |
|:---:|:---:|
| (**a**) | (**b**) |

**Figure 12.** (**a**) Enthalpies of hydrate formation in units kJ/mole hydrate. Red solid is for pure $H_2S$ hydrate and black solid is for pure $CH_4$ hydrate. Dashed blue is for hydrate formed from a mixture of 70% $CO_2$, 5% $CH_4$, 5% $C_2H_6$, and 20% $N_2$. Dashed red is for hydrate formed from a mixture of 70% $CO_2$, 5% $CH_4$, 2% $C_2H_6$, and 23% $N_2$. (**b**) Enthalpies of hydrate formation in units kJ/guest molecule in hydrate. Red solid is for pure $H_2S$ hydrate and black solid is for pure $CH_4$ hydrate. Dashed red is for hydrate formed from a mixture of 70% $CO_2$, 5% $CH_4$, 5% $C_2H_6$, and 20% $N_2$. Dashed red is for hydrate formed from a mixture of 70% $CO_2$, 5% $CH_4$, 2% $C_2H_6$, and 23% $N_2$. Black square is experimental data for pure $CH_4$ hydrate from Handa et al. [53]. Black pentagram is experimental data for pure $CH_4$ hydrate from Rueff et al. [54]. Red circle is experimental data for pure $H_2S$ hydrate from Yoon et al. [55]. Black diamond is experimental data for pure $CH_4$ hydrate from Lievois et al. [56]. Black circle is experimental data for pure $CH_4$ hydrate from Handa et al. [53].

Based on the enthalpies of hydrate formation for pure $CH_4$ hydrate, and this for pure $H_2S$ hydrate, it seems obvious that addition of 1 to 3 mole% $H_2S$ will imply some limited shifts of the black curves for the pure $CH_4$ hydrate in Figure 12a,b. It is, therefore, clear that also criterion 3 will be satisfied for the mixtures with added $C_2H_6$, except from some very limited regions, according to criterion 1. This is not even critical, as mentioned above.

The mechanism is on pore scale and involves the exchange of heat across a liquid film separating the in situ hydrate from the new hydrate formed from the injection gas. It is, therefore, expected that criterion 4 is satisfied. Without detailed data that can provide a basis for setting up realistic heat transfer models, it is impossible to quantify criterion 4 in more detail. Heat transfer through liquid water is 2–3 orders of magnitude faster than mass transport [44,56], and it is not expected that criterion 4 represents a kinetic bottleneck in $CO_2/CH_4$ swapping when the injection gas mixture is adjusted as discussed above, and a proper second additive is added. The main purpose of the second additive is to prevent the formation of new hydrate from the injection gas to create blocking hydrate films that slow down the $CO_2/CH_4$ swapping.

## 4. Discussion

There are many frequent misunderstandings related to the use of $CO_2$ for producing $CH_4$ hydrates, while at the same time storing $CO_2$ safely as hydrate. One of these is that the kinetics is slow. The use of pure $CO_2$ will lead to very slow conversion for the simple reason that $CO_2$ hydrate forms instantly on a macroscopic time scale (seconds and up). The instant formation of blocking $CO_2$ hydrate films lead to the slow progress of $CO_2/CH_4$ swapping. Hydrate nucleation is a nano-scale process in time and volume. It is fundamentally different from hydrate induction times. Hydrate nucleation is the initial stage of hydrate formation needed to create a hydrate particle that is large enough to enter steady hydrate growth. Physically, it means that the benefits of the hydrate formation Gibbs free energy change has to overcome the penalty of pushing away the surrounding

original phases. If there are no external forces related to flow, the steady growth stage may be very slow since hydrate formers have to diffuse through the solid hydrate film, or liquid water molecules are transported to the gas side of the hydrate film [36]. Induction time can be interpreted as the time needed to reach massive hydrate growth. See Kvamme et al. [36] for the prediction of hydrate induction times for $CH_4$ hydrate, as well as $CO_2$ hydrate. The nano- to mesoscale nature of hydrate nucleation and growth has been studied in several papers, and there is no need for separate calculations in this paper. Interested readers are directed to references [35–39] for examples of heterogeneous and homogeneous hydrate nucleation for $CH_4$ and $CO_2$ hydrates. More advanced models, such as, for instance, Phase Filed Theory (PFT), are tools that can shed a deeper light on more complex nucleation and growth processes, such as, for instance, situations in which lack of mass for further growth leads to growth of more stable cores (more negative Gibbs free energy) at the cost of decays of less stable hydrate particles. See, for instance, references [43–48] and references in those theses and papers.

In summary, there is a need to focus on the role of additives. Adding $N_2$ and other gases, as illustrated here, has two purposes. The first purpose is to increase injection gas permeability relative to pure $CO_2$. The other purpose has been illustrated in more detail here. Manipulations of the injection gas composition can satisfy a set of criterions which thermodynamically ensure that $CO_2/CH_4$ swapping is feasible.

A second set of additives is needed. These are additives that are active on the injection gas/pore water interface, with a primary purpose of reducing blocking hydrate films that will slow down $CO_2/CH_4$ swapping. Even small alcohols, such as methanol and ethanol [47,48], have surfactant effects when water containing these alcohols is exposed to a non-polar phase, such as $CH_4$ or $CO_2/N_2/CH_4/C_2H_6$ mixtures. The reason is the low partial charge on the methyl group in methanol relative to the size of the methyl group. For ethanol, the outer methyl group is practically non-polar. The advantage of these small alcohols is that they move very fast, along with water in the interface, i.e., water and these small alcohols have similar diffusivity coefficients in the interface [57–60]. Small surfactants, on the other hand, will remain at the $CO_2$/water interface and are also needed. The addition of small amounts of mixtures of alcohol and surfactant will actually speed up the formation [57–60] of hydrate formation from the injection gas, while at the same time assisting in keeping the interface free of blocking hydrate films.

Another misinterpretation is that $CO_2$ hydrate is less stable than $CH_4$ beyond 284.14 K (the phase transition to a higher density). This is thermodynamically incorrect. Hydrate stability is determined by the level of Gibbs free relative to the stability of the original phases. Since we use a uniform reference state for all components in all phases (ideal gas pure component), the Gibbs free energy of different hydrates can also be compared. Temperature and pressure are independent thermodynamic variables, and in terms of the feasibility of $CO_2/CH_4$ swapping, these variables are only used in the evaluation of regions where the different types of hydrate can exist. Yet, another set of misinterpretations is related to the fact that hydrates in natural sediments cannot reach thermodynamic equilibrium. The reason is as simple and as complex as one wants to look at it. The simple way to look at it stems from old-time hydrate experiments from more than eight decades ago. It was well known at that time that only one independent thermodynamic variable could be fixed in order to measure the hydrate equilibrium curve for a single hydrate former. Despite this fact related to the Gibbs phase rule, the same curves are also used as equilibrium curves in natural settings, where two independent variables (temperature and pressure) are fixed. The complexity is substantially greater than that. Some of these aspects are discussed by Kvamme et al. [30] and Kvamme [31,32]. Even though the Gibbs phase rule can be used for hydrate systems, some care is needed since interfaces play a substantial role. Examples are the role of mineral surfaces [30] in hydrate nucleation, and selective adsorption on the liquid water surface prior to hydrate formation [14].

Also note that mineral/water/hydrate gas interactions are not discussed explicitly in this work. The reason is that the primary impact of minerals does not affect the thermodynamics related to the four criterions.

We have discussed these effects in a variety of papers directly related to hydrates, and also in papers related to various aspects of $CO_2$ storage in aquifers. Rather than adding too many references, we will limits ourselves to a few references, some of the main effects of mineral/fluid interactions, and impacts for hydrate, while explaining why these aspects are not very important for this work.

Thermodynamically, it is impossible for hydrate to stick to mineral surfaces. Adsorbed water chemical potential is simply far too low compared to hydrate chemical potential and "bulk" chemical potential. This can be seen from experimentally sampled adsorption of water on minerals. Three-times liquid water density is not unusual. Converting this into distribution functions and connection to the canonical partition function provides the link to Helmholtz free energy. Molecular Dynamics simulations with utilization of modern sampling methods for chemical potentials [61] also provide a link between sampled structures of adsorbed water on mineral surfaces and water chemical potential. The very low chemical potential of adsorbed water is impossible for hydrate water. Any hydrate approaching a mineral surface will dissociate because the water molecules will thermodynamically prefer to transfer from hydrate to adsorbed on a mineral surface. The journal papers enclosed in Olsen's thesis [61] also includes references to experimental structure data. Comparing the geometrical distribution of atomistic charges in mineral surfaces and the geometrical distribution of partial charges in hydrate water will also illustrate that it is not electrostatically beneficial for hydrate to touch mineral surfaces directly. Hydrate can, however, be "bridged" to hydrate by two interfaces. The interface between water adsorbed directly on a mineral surface and "bulk water" is in the order of five water layers (around 1.6 nm) [61]. The hydrate/liquid water interface is in the order of 1.2 nm thick [43,47,48]. A minimum distance between minerals and hydrate of roughly 3 nm is, therefore, controlled by mineral/water and hydrate/water interfaces. This minimum distance is normally much larger due to molecular transport effects (diffusion) and other effects imposed by dynamics (many reasons) on the hydrodynamic level. Even in the most closed reservoirs in permafrost, it is therefore hard to find any hydrates with a saturation higher than 85% of pore volume.

The "catalytic" effect of mineral surfaces is extremely important in the nucleation process. Hydrate nucleation is a nano-scale process in time and volume [35–38]. From a macro perspective (seconds and up), hydrate nucleation is "instant". For this classical thermodynamic analysis, the water/gas/hydrate/mineral interactions are not important as long as we know that there is enough free water in the pores to create a new hydrate from the injection gas mixture. Hydrate nucleation has been discussed in several of our papers and calculations presented. The critical issue is the formation of blocking hydrate films. These are the reasons for the second additive. The actual phenomena of relevance are not the physically well-defined hydrate nucleation and hydrate growth. Hydrate induction can be expressed as "onset of massive growth". Induction times have been discussed by Kvamme et al. [36], while some aspects of secondary hydrate nucleation towards mineral surfaces have been discussed in reference [30]. Injection of pure $CO_2$ is not relevant for reasons discussed here (low injection permeability and formation of hydrate-blocking films). The quadrupole moment of $CO_2$ will result in some adsorption of $CO_2$ directly on mineral surfaces [61–63] (and papers included in these theses). This opens up several routes to hydrate nucleation that include adsorbed $CO_2$. Adsorbed $CO_2$ and surrounding adsorbed water can restructure, release from the adsorbed state, and form hydrate outside of mineral surfaces. The dynamics of water and $CO_2$ adsorption can results in hydrate nucleation from adsorbed $CO_2$ and water outside mineral surfaces. $CO_2$ might be trapped in structured water outside mineral surface and can nucleate there, similar to $CH_4$ trapped in structured water. All these different options have to be investigated further through a variety of methods that range from nano-scale methods, such as Quantum Mechanics and Molecular

Dynamics simulations, to meso-scale methods, such as Phase Field Theory [43–48]. The first type of additive (primarily $N_2$ in this study) may not affect the $CO_2$ adsorption, except for the effect of $N_2$ on the $CO_2$ chemical potential in the fluid. The exception is $H_2S$. Surfactant additives may, however, affect the adsorption of guest molecules on mineral surfaces. Finally, it is all connected to the flux of the injection gas through the $CH_4$ hydrate-filled sediments. As such, the effect of the surfactant additive is also critical in determining the nucleation flux of $CO_2$ towards mineral surfaces. Even though nucleation is "instant" on a macro level, the number of active hydrate cores will also affect the hydrate induction times.

As mentioned in the introduction, fractures that bring seawater into $CH_4$-filled sediments lead to hydrate dissociation. Thermodynamically, there is nothing very special about this. It is just a consequence of the multi-dimensional thermodynamic dependency of hydrate stability, as given by Equations (4) and (5). In particular, if the chemical potential for a guest molecule is lower and seawater is surrounding the hydrate, then the hydrate is not unconditionally stable, even if the temperature and pressure are inside the hydrate formation region of conditions. Before presenting some relevant numerical examples, it is worthwhile to return to degrees of freedom and the Gibbs phase rule. As discussed above, there is one degree of freedom with water and one hydrate formerly distributed for the gas, liquid, water, and hydrate. Fixing two independent variables obviously then imposes a mathematically over-determined system. Going back to the basis for the Gibbs phase rule, then it is simply a balance between the number of independent thermodynamic variables minus the constraints on these [30,31,64]. Constraints are mass conservation and equilibrium conditions. If we just assume that mechanical equilibrium establishes locally (same pressures in all phases), and also thermal equilibrium establishes (same temperatures in all phases). The number of independent thermodynamic variables in the three phases boils down to six: the mole-fractions of water and one guest in each of the three phases. The conservation of mole-fractions in each phase gives three conservation laws. There are four independent equilibrium conditions for the chemical potentials of water and guest in different phases. Obviously, the conservation laws have to be met, and that leaves a net of three independent variables and four constraints on chemical potentials for water and guest in different phases. It is, therefore, impossible mathematically to satisfy these four constraints, and the chemical potentials cannot be the same for the two components in the three phases. The phase distributions and associated phase compositions are given by the local minimum Gibbs free energy, according to Equation (4). The minimum mole-fractions of $CO_2$ in surrounding pure water needed to keep $CO_2$ hydrate stable is plotted in Figure 13a (black), along with the liquid water solubility (red). The corresponding water chemical potentials are plotted in Figure 13b. Similar plots for seawater do not differ very much, and the reason for plotting these for pure water is simply because there are more available experimental data for pure water, in case readers want to compare the results in these plots with available data from the literature. Water solubilities in $CO_2$ fluid are less important in the content of this manuscript and are not plotted here. Similar plots for $CH_4$ are given in Figure 14. The black curve in Figure 14a is the reason that hydrate dissociates due to incoming seawater through fractures connecting to the seafloor. Fairly many offshore hydrate systems are in a dynamic balance between formation of new hydrates from upcoming gas and dissociation towards incoming seawater (almost no $CH_4$) from the top of the hydrate stability zone. Even fluxes from conventional hydrocarbon systems may form hydrate when contacting seawater at the seafloor, if the conditions of the temperature and pressure are favorable. These hydrates are, however, in a dynamic balance between the hydrate dissociation in the $CH_4$ chemical potential gradients between the $CH_4$ chemical potential in the hydrate and the chemical potentials in the outside water (almost infinite dilution).

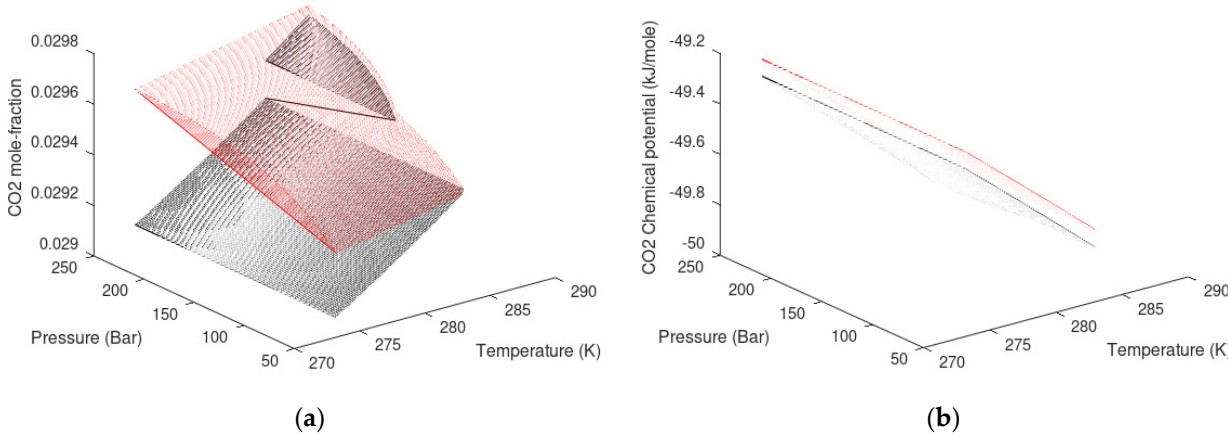

**Figure 13.** (**a**) Minimum mole-fraction of $CO_2$ in surrounding water needed to keep $CO_2$ hydrate stable (black). Solubility of $CO_2$ in water (red). (**b**) Chemical potential for water corresponding to mole-fractions $CO_2$ in water for minimum mole-fractions $CO_2$ in surrounding water needed to keep hydrate stable (black) and chemical potentials for water in $CO_2$ solubility mole-fractions (red).

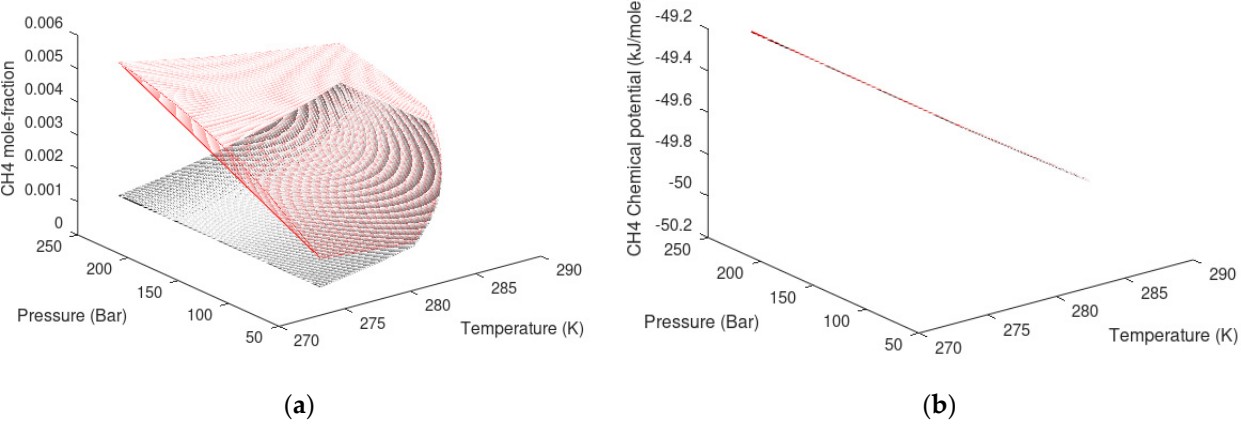

**Figure 14.** (**a**) Minimum mole-fraction of $CH_4$ in surrounding water needed to keep $CH_4$ hydrate stable (black). Solubility of $CH_4$ in water (red). (**b**) Chemical potential for water corresponding to mole-fractions $CH_4$ in water for minimum mole-fractions $CH_4$ in surrounding water needed to keep hydrate stable (black) and chemical potentials for water in $CH_4$ solubility mole-fractions (red).

Although the primary focus of this work is the storage of $CO_2$, and the associated release of $CH_4$ is an economic benefit, it is worthwhile to mention an interesting approach for thermal stimulation using emissions from underwater mud volcanos [65]. Actually, if the energy from these underwater mud volcanos can be directed and distributed in a fashion such that the creation of a new hydrate from the injection gas is not inhibited by this extra thermal stimulation, it will absolutely be very beneficial.

Another interesting study [66] focuses on hydrate dissociation zones and contains laboratory experiments as part of the analysis of Black Sea hydrates. This study is, however, not relevant for the characteristics of hydrate dissociation in this paper. The creation of new hydrate from injection gas and the dissociation of in situ hydrates in the same pores results in very specific patterns of $CH_4$ hydrate dissociation. The paper by Bazaluk et al. [66] will, however, be more important for other types of thermal stimulation related to a more specific focus on optimising the $CH_4$ production.

Finally, it is important to emphasize that there is no need for new technology. Injection of $CO_2$ for aquifer storage has been a daily routine since 1996, and capturing released $CH_4$ after the swapping is basically standard petroleum technology. As discussed above, the important challenge is to design efficient injection gas mixtures and efficient surfactant mixtures that can reduce blocking hydrate films.

Black Sea hydrates are mentioned in the title because they have been used as examples for illustration of the method, and because they represent a very interesting European possibility. The method is, however, general, and some relevant studies for the near future may include Gulf of Mexico hydrates since the Department of Energy (DOE) has a huge budget for safe storage of $CO_2$, which can be relevant for $CO_2$/$CH_4$ swapping. China has a huge focus on hydrate energy, and it would be interesting to examine some of the most relevant hydrate reservoirs of offshore China using the proposed method. Yet another possibility in China lies in the permafrost onshore hydrates. The paper by Li et al. [67] provides some information and further references that can serve as a starting point for the evaluation of the feasibility for $CO_2$/$CH_4$ swapping in China permafrost hydrates. These regions are very interesting as a basis for a combination with steam cracking since that will generate a standalone energy generation facility, which can be used actively to develop regions of China that would otherwise be expensive and difficult to develop. Based on preliminary numbers from [67], the potential deliverance of $H_2$ from such a plant can deliver huge amounts of energy for local development and export. In general, the list of potential worldwide projects that can be developed can be very long.

## 5. Conclusions

$CO_2$/$CH_4$ swapping has been studied experimentally in several types and sizes of experimental set-up. Even a pilot plant study has been conducted in Alaska. A systematic limitation in these studies is the lack of a thermodynamic analysis prior to the "design" of the injection mixture. Typically, the process is a trial-and-error method. In this work, we have proposed and utilized a systematic thermodynamic method for analyzing the feasibility of $CO_2$/$CH_4$ swapping in terms of the fundamental thermodynamic laws. The use of fundamental thermodynamic laws and a uniform reference system that permits thermodynamic comparison of different hydrate phases, as well as other possible phases, is unique and represents the novelty of this work. Utilization of the proposed method can save money in the planning of expensive experiments and pilots through the theoretical design of suitable injection mixtures, rather than expensive trial-and-error procedures. We illustrated the method using information from several sites in the Black Sea. It is known that there is some $H_2S$ in some of the sites, and it appears that all the conditions of the bottom of hydrate stabilities fall in between zero $H_2S$ and maximum 3 mole% $H_2S$. One challenge with using $H_2S$ is the jump in hydrate stability limit pressure at 283.14 K, when $CO_2$ undergoes a density increase. For temperatures above this phase transition temperature, the hydrate stability limit pressures for $CO_2$ are higher than that of $CH_4$ hydrate. Temperature and pressure are, however, independent thermodynamic variables and are not measures of hydrate stability. Hydrate stabilities are determined by the combined first and second laws of thermodynamics, as expressed by Gibbs free energy. Hydrates of $CO_2$ and the mixtures of $CO_2$ with other components examined in this study are all more stable than in situ hydrates form the Black Sea. Regions of hydrate existence in terms of temperature and pressure (criterion 1) are not necessarily critical, as long as temperatures and pressures are in regions where both in situ hydrate can exist and hydrate from injection gas ($CO_2$ with additives) can exist. The reason is that the new hydrate from injection gas will form from free-pore water, and the released heat will dissociate the in situ hydrate. The two hydrates will, therefore, not be in direct contact and, as such, not be in competition. The second criteria related to the combined first and second laws also has a second criteria. Unconditional hydrate stability also implies that all gradients of the Gibbs free energy need to result in hydrate stability. One consequence of this is that the hydrate water chemical potential has to be lower than the liquid water chemical potential. This criterion puts even

more strict restrictions on the maximum amounts of $N_2$ that can be added. More than 30 mole% $N_2$ is not likely to result in the formation of new hydrate from injection gas. A third criterion is related to the first law of thermodynamics and the energy balance between heat released from the formation of hydrate from injection gas relative to what is needed to dissociate in situ hydrates. Only a limited mixture for injection gas has been examined as illustration. A mixture of 70 mole% $CO_2$, 20 mole% $N_2$, 5 mole% $CH_4$, and 5 mole% $C_2H_6$ will satisfy most of the actual situations and only leave an extremely small region of temperature and pressure, for which an in situ system consisting of 97 mole% $CH_4$ and 3 mole% $H_2S$ would have lower hydrate stability limit pressure than the stability limit pressure that hydrate stability limits pressure for the injection gas. A fourth criterion is related to the second law of thermodynamics and puts demands on the level of temperature applied to dissociate the in situ hydrate. Since this is a pore-scale mechanism and hydrate formation that delivers dissociation heat is very close to the in situ hydrate, we expect that this criterion is met as long as the injection gas is able to create a new hydrate. Adjustment of the injection gas to increase injection gas permeability and to meet thermodynamic criterions are one side of the challenge. The other side of the challenge is to reduce blocking hydrate films from injection gas hydrate formation. For this, addition of small amounts of surfactant mixture is needed.

**Author Contributions:** Conceptualization, methodology, and investigation, B.K. and A.V.; software, B.K.; data curation, A.V.; draft preparation, review, editing, and visualization, B.K. and A.V.; project administration, A.V. All authors have read and agreed to the published version of the manuscript.

**Funding:** This research was funded by the Bulgarian Science Fund project KP-06-OPR04/7 GEO-Hydrate "Geothermal evolution of marine gas hydrate deposits—Danube paleodelta, Black Sea" (2018–2023) and the European Union project 101000518 DOORS: Developing Optimal and Open Research Support for the Black Sea (2021–2025).

**Data Availability Statement:** Not applicable.

**Acknowledgments:** We would like to thank the reviewers for the time and effort necessary to review the manuscript and improve its quality. Bjørn Kvamme is grateful for financial support through 111 Project (No: D21025), National Key Research and Development Program (No: 2019YFC0312300), National Natural Science Foundation Item of China (No: U20B6005-05, 51874252 and 5177041544), Open Fund Project of State Key Laboratory of Oil and Gas Reservoir Geology and Exploitation (No: PLN$_2$021-02 and PLN$_2$021-03).

**Conflicts of Interest:** The authors declare no conflict of interest.

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
