# Peer review of "Thermodynamic Feasibility of the Black Sea CH4 Hydrate Replacement by CO2 Hydrate"

_energies, doi:10.3390/en16031223_

Round 1

Reviewer 1 Report

The authors investigated the thermodynamic feasibility of the replacement of CH4 hydrate with hydrate formed by CO2 and other gas. Two basic logic behind the feasibility evaluation are, to be able to achieve the successful replacement, the phase diagram of hydrate formed by injection gas mixture has to be below the phase diagram of in-situ hydrate (CH4) hydrate and the Gibbs free energy change during transition from CH4 hydrate to the other one formed by injection gas mixture must be negative. Authors also optimized the formula of injection gas mixture using thermodynamic criteria. However, the replacement technology has been proposed for a long time and widespread studies have been conducted by researchers. Some groups also reported the injection of CO2 and N2 mixture as a replacement medium of CH4 hydrate. Therefore, this article is to be improved in terms of novelty. Besides, the principles outlined in this article, for instance, the thermodynamic laws and the Gibbs free energy criteria, are all quite fundamental, making this paper not an advanced research article. Moreover, there are several major problems in this manuscript:

1.     The readability of this manuscript is not very good. Authors should use comma or other punctuations to make the article clearer and smoother. Some sentences are incomplete and ambiguous. For example, row 271, “The second term is denoted as shaft work because”. Because what?

2.     Authors stated that mixing nitrogen or air with CO2 is beneficial for the enhancement of gas permeability. Why? This is the reason why nitrogen was added and needs clarification.

3.     Authors stated the phase transition of CO2 and a density increase slightly above 284.14 K. To the best of the reviewer’s knowledge, remains a gaseous state at this temperature if the pressure is atmospheric pressure. The density of CO2 decreases with increasing temperature.

4.     For the abbreviation that appears for the first time, full name is required. For example, DSDP in figure 1. This is quite important for non-expert readers. Some definitions, such as induction time, lack accuracy.

5.     The subsection “Black Sea hydrate” is too long. Some less important information can be transferred to the Supplementary materials.

6.     How did authors calculate and plot figure 5a and 5b? Also, for the legend in figure 5, “from 99 mole% CH4 and 2 mole% H2S (green) and 99 mole% CH4 and 3 mole% H2S (red).” The total composition is not 100%. This should be double checked.

7.     Many typos throughout the manuscripts, including but not limited to the following two examples. Row 329, “un” should be “in”; row 330 “form” should be “from”;

8.     All equations should be referred in the text. There is no reference for equation 14.

9.     There are too many mistakes in figures and their captions.

9.1  There is no caption for figure 6 c and d.

9.2  Row 385. Figure 1 is the topographic map. I cannot find figure 1(c) and 1 (d).

9.3  Unit of y axes in figure 8 a) is wrong. Is the unit of mole fraction kJ/mol?

9.4  There is only one red dashed curve in figure 8 b) but the caption stated there are two.

9.5  Row 513-514. Which composition does red curve represent after all?

9.6  Row 527. The composition of 75 % CO2, 2 % H2S, 3% CH4 and 23% N2 made a total of 103%.

9.7  Why Figure 12 caption showed so many “dashed red”?

10.  The discussion part looks inappropriate. Discussion part should be based on the authors’ results not the correction of misinterpretations. Authors made statements about nucleation kinetics and surfactant here, but these are irrelevant to the topic of this paper.

11.  The conclusion part is too long. Many sentences are not necessary. Keep the conclusion as brief and highlighted as possible.

Author Response

Plz see enclosed file

Reviewer 2 Report

The present manuscript describes the investigation of the hot topic. Moreover, it is critically important in terms of russian war in Ukraine because of the necessity to investigate and develop alternative sources of energy.

The content of the manuscript is similar to that of a feasibility study. The knowledge contained here may be useful for engineers, students, and scientists, searching for any knowledge related to gas hydrate replacement by carbon dioxide, which is the most important value of the manuscript.

A manuscript has a practical application and also provides important theoretical for the next studies.

The paper can be accepted for publication after providing the corrections mentioned below.

Point 1. The abstract section sounds unclear. The abstract should follow the MDPI style of structured abstracts:

- Background (place the question addressed in a broad context and highlight the purpose of the study);

- Methods (describe briefly the main methods);

- Results (summarize the article's main findings);

- Conclusion (indicate the main conclusions or interpretations).

Point 2. In the Introduction section, an enhanced literature review is required. For this study, the authors have used only 11 literature sources. It seems insufficient for such type of research.

Please consider the work of authors from Ukraine in the research field:

Klymenko, V., Ovetskyi, S., Martynenko, V., Vytiaz, O., & Uhrynovskyi, A. (2022). An alternative method of methane production from deposits of subaquatic gas hydrates. Mining of Mineral Deposits, 16(3), 11-17. https://doi.org/10.33271/mining16.03.011

Bazaluk, O., Sai, K., Lozynskyi, V., Petlovanyi, M., & Saik, P. (2021). Research into Dissociation Zones of Gas Hydrate Deposits with a Heterogeneous Structure in the Black Sea. Energies, 14(5), 1345. https://doi.org/10.3390/en14051345

Point 3. It will be great if the authors show some description in context – Why it is important to conduct this study? Can the expected result be used or implemented within other gas hydrate placement conditions? If yes, then how? What limitations?

Point 4. It is difficult to understand figures without legends. Definitely, colors are indicated below the figures, but it will be better to see legends on the figures.

Point 5. Please provide a short description of further research.

Point 6. The novelty of the paper must be highlighted in the conclusions section.

Point 7. In general, I must admit that a very good study was performed, and I will recommend your paper for publication after careful revision.

Author Response

See enclosed file

Reviewer 3 Report

This work presents an evaluation on the feasibility of CO2 injection into methane hydrate bearing sediments. The topic is interesting to me; however, the manuscript cannot be publishable at its current form unless the authors sufficiently address the following comments:

1.     Please revise the title of this study to clarify to the reader that this work deals with the thermodynamics aspects of the CH4 replacement with CO2 in hydrate reservoirs.

2.     I can’t find the abstract consistent. In particular, the first few sentences are some detailed information, probably pasted from somewhere else. Please carefully revise it.

3.     Generally, I find the introduction section interesting; however, it’s not well-structured, the text is not scientific enough, and several statements in introduction are not well-supported. For instance,

-        “Sealing (shale, clay) integrity has been verified through millions of years of trapping the natural gas hydrate.”. Is there any reference/evidence to support this statement? The submarine sediments with potential for hydrate formation are usually unconsolidated, so, would it be possible to have shale formations in the shallow sediments?

-        “(for the Black Sea GHDs in average ~2%; see below).” What does “below” refer to?

-        What is the connectivity of the CO2 injection into aquifer in the second paragraph to the key messages this study would convey?

-        …

Please carefully revise this section, modify its structure, and add proper references to support your statements.

4.     Reference 10 is not published yet. If not, I think it should be removed.

5.     Figure 1 is interesting however it’s quite blurred and the legend is very difficult to read. The BSR areas are difficult to locate. Please enlarge the figure and the caption and use the high-quality image; please do the same for figures 2 to 4, too.

6.     “G” has been used for both geothermal gradient (Table 1) and fee Gibbs energy (figure 5). Am I right? I also see “g” is used for free Gibbs energy in equation 11 to 13.

7.     “Researchers that want to reproduce the results presented here will therefore benefit from also reading the original papers describing the different aspects of the model.” What papers does this statement refer to? Please cite them.

8.     “With 2 components (water and CH4) distributed over 3 phases (water, gas hydrate) …” What is the third phase?

9.     “Without detailed composition information on the hydrates in table we use …” Which table is referred to?

10.  I understand this work investigates the feasibility of CH4 replacement by CO2 from a thermodynamic perspective; however, I was wondering if the authors could elaborate regarding the real conditions where the rock and fluid interactions become important too?

Author Response

See enclosed answers and also the revised version attached from page 10

Round 2

Reviewer 1 Report

The authors have satisfactorily addressed the previous comments.

Author Response

Thank you very much. 

Reviewer 2 Report

Dear author, You have done a good revision.
I hereby recommend your paper for publication.

Congratulations.

Author Response

Thank you very much